 SciPost Phys. Lect. Notes 52 (2022)

# Dark atoms and composite dark matter

**James M. Cline***

McGill University, Dept. of Physics, Montréal, Québec, Canada

⋆ [jcline@physics.mcgill.ca](mailto:jcline@physics.mcgill.ca)

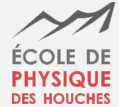

*Part of the Dark Matter*
*Session 118 of the Les Houches School, July 2021*
*published in the Les Houches Lecture Notes Series*

## Abstract

I selectively review the theoretical properties and observational limits pertaining to dark atoms, as well as composite dark matter candidates bound by a confining gauge interaction: dark glueballs, glueballinos, mesons and baryons. Emphasis is given to cosmological, direct and indirect detection constraints. Lectures given at Les Houches Summer School 2021: Dark Matter.

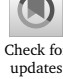
doi:10.21468/SciPostPhysLectNotes.52

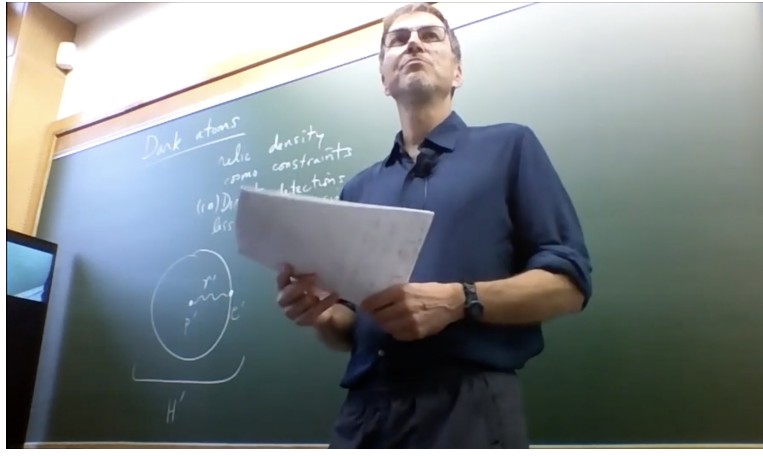

# 1 Introduction

One of the earliest theoretical postulates for dark matter (DM), not to mention the now-popular framework of hidden sectors, was a mirror of the standard model (SM) [1–3]. It provided two examples of stable composite particles that could serve as dark matter: atoms and baryons (nuclei). Other early DM candidates included magnetic monopoles, axions, massive neutrinos, sneutrinos and photinos [4]. Supersymmetric neutralinos seemed to enjoy a favored status for many years. The idea of a hidden sector with complex structure—gauge interactions and DM multiplets—was revitalized outside of the mirror context by Refs. [5, 6], inspired by cosmic ray anomalies [7]. Around the same time the "hidden valley" paradigm [8] of a new confining gauge was proposed, mostly with signals for the Large Hadron Collider in mind, but also with the awareness that a stable bound state could serve as the DM.

In these lectures we review the various possibilities for dark matter in the form of bound states, either in the case of a U(1)′ dark gauge group, leading to dark atoms, or of a confining interaction, which could give rise to dark glueballs, mesons or baryons. Because atoms and baryons also exist in the visible sector, it is reasonable to suppose they get their relic density in a similar way as for visible matter—*i.e.*, we don't know! In other words, they could be asymmetric DM [9], the origin of whose asymmetry remains to be explained. Beyond their relic density, many interesting aspects can be addressed, in terms of direct and indirect signals.

For some varieties of composite DM, it is not possible to have an asymmetry, in which case a calculation of the relic abundance is definitely called for. A confining phase transition can make the freezeout process more complex than in the standard thermal freezeout picture, often depending on the relative temperatures of the two sectors. There is the important model-dependent issue of which portals, if any, exist between the hidden and visible sectors. Of course gravity always exists, and even it can play a role, as we will see for glueballs. Although in these lectures I will focus on hidden sectors, it is interesting to note that DM could be a $\sim 25\,\text{TeV}$ composite state of gluino-like fermions bound by QCD [10, 11].

# 2 Dark atoms

The simplest dark atom model outside of the mirror framework was studied in Refs. [12, 13]. It consists of a dark electron, proton and photon, $e'$, $p'$ and $\gamma'$ respectively, with a coupling strength of $\alpha' = g'^2/4\pi$. In its minimal version, the only other fundamental parameters needed are the masses $m_{e'}$ and $m_{p'}$. Later we will consider the consequences of also including a photon mass $m_{\gamma'}$ and kinetic mixing $\epsilon$ with the SM hypercharge. An important derived quantity is the binding energy of the dark $H'$ atom,

$$B_{H'} = \frac{\alpha'^2}{2}\mu_{H'} = \frac{\alpha'^2}{2}\frac{m_{e'}m_{p'}}{m_{e'}+m_{p'}} \cong \frac{\alpha'^2}{2}m_{H'}\frac{R}{(1+R)^2}\,, \tag{1}$$

where $\mu_{H'}$ is the reduced mass and $R = m_{p'}/m_{e'} > 1$. (Without loss of generality one can assume that $R \geq 1$ since the sign of the U(1)′ charge is arbitrary.) The mass of the $H'$ atom is therefore $m_{H'} = m_{e'} + m_{p'} - B_{H'}$, which we usually approximate as $m_{e'} + m_{p'}$, unless one is interested in the regime of strong coupling. The condition $m_{H'} > 0$ implies the weak constraint

$$\alpha' < \sqrt{2}(1+R)\,. \tag{2}$$

If it was violated, the mass of the atom is of course not negative; rather the perturbative calculation (and possibly the nonrelativistic approximation) is breaking down.

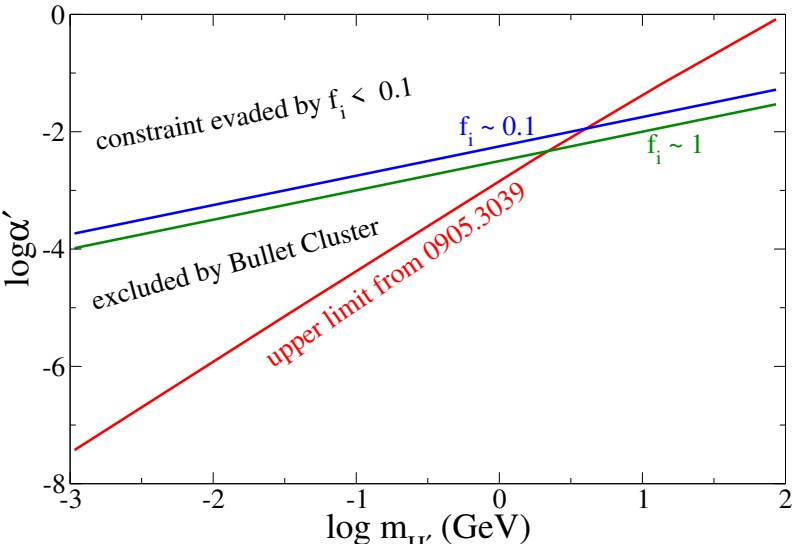

Figure 1: Region of dark atom parameter space excluded by Bullet Cluster bounds for models with $R = 1$ ($m_{e'} = m_{p'}$).

## 2.1 Cosmological evolution

The cosmology of dark atom formation was initially worked out in Ref. [12], and later in more detail by Ref. [14]. In the absence of a portal between the two sectors, the dark sector will generally have a different temperature $T'$ from the visible one $T$, and their ratio $\xi = T'/T$ can evolve with time. One could imagine some initial ratio $\xi_i$ that is set by the relative efficiency of reheating to the two sectors after inflation.

At temperatures $T' \sim m_{p'}/20$ and $T' \sim m_{e'}/20$ respectively, the symmetric components of $p'$ and $e'$ freeze out through the annihilations $p'\bar{p}' \to \gamma'\gamma'$ and $e'\bar{e}' \to \gamma'\gamma'$. This is followed by recombination at $T' \lesssim B_{H'}$, below the binding energy due to the small concentration of baryons relative to photons. If $\xi_i \ll 1$, then all of these events occur when $T$ is significantly higher in the visible sector, *i.e.*, at relatively early times compared to SM recombination.

However it need not be the case that $\xi_i \ll 1$. The dark photons are extra radiation species contributing to the Hubble expansion, conventionally parametrized as extra neutrino species,

$$\Delta N_{\text{eff}} = 4/7 \left( \frac{11}{4} \right)^{4/3} g'_* \xi^4 < 0.45 \,, \tag{3}$$

where $g'_* = 2$ if $\gamma'$ is the only dark radiation species, $(11/4)^{1/3} = T_\gamma/T_\nu$ accounts for the differential heating of photons versus neutrinos after freezeout of the weak interactions, and $4/7 = \frac{1}{2}(\rho_\gamma/\rho_\nu)$ for a single $\nu$ species. The upper bound is from *Planck* cosmic microwave background (CMB) constraints [15]. Solving (3) one finds the modest constraint

$$\xi = \frac{T'}{T} < 0.57 \tag{4}$$

at late times. This would naturally result even if $\xi_i = 1$ after inflation, if the two sectors remained decoupled, due to the much larger entropy in the visible sector [16].

An important quantity is the ionization fraction $f_i = n_{e'}/n_{H'}$ after recombination, the number of free $e'$ particles per dark atom. It is determined by solving the appropriate Boltzmann equations describing recombination. Ref. [17] made an analytic fit to the numerical results of [12],

$$f_i \cong \min \left[ 1, \, 10^{-10} \xi \, \alpha'^{-4} R^{-1} \left( \frac{m_{H'}}{\text{GeV}} \right)^2 \right] \,. \tag{5}$$

Independently, Ref. [14] arrived at the estimate

$$f_i \sim 2 \times 10^{-16} \, \xi \, \alpha'^{-6} \left(\frac{m_{H'}}{\text{GeV}}\right) \left(\frac{B_{H'}}{\text{keV}}\right),$$ (6)

which is compatible with (5) (using Eq. (1)) for $R \gg 1$.

A large ionization fraction would be problematic because of the strong Coulomb interactions between ions, in violation of Bullet Cluster constraints on DM self-interactions [18,19]. The resulting bound on $\alpha'$ was derived in Ref. [20] assuming that $f_i = R = 1$,

$$\alpha' < 10^{-2.9} \left(\frac{m_{H'}}{\text{GeV}}\right)^{1.5},$$ (7)

which is my fit to their numerical result (red line, Fig. 1). However one should combine this with the estimate (5) of $f_i$ as a function of $m_{H'}$ and $\alpha'$ to see what region of parameter space is actually excluded. I have done this exercise in Fig. 1. The actual excluded region is the triangle at lower $m_{H'}$ between the red and blue lines. The blue line represents $f_i = 0.1$, where the Bullet Cluster bounds would be evaded.

More stringent bounds on $\alpha'$, by a factor of $10^4$, have been derived on the basis of observed DM halos with elliptical rather than spherical morphology, since DM self-interactions would tend to erase the ellipticity [21]. However, subsequent analyses indicated that the ellipticity bound is not as stringent as originally thought, but rather of the same order as the Bullet Cluster constraint [22].[1]

Even if there is no significant ionization at early times, dark atoms can reionize during structure formation, by shock heating as they concentrate within galactic halos. This causes the atoms to heat to the virial temperature, which scales with redshift $z$ as [24]

$$T_{\text{vir}} \sim G \, M_{\text{halo}}^{2/3} \, \rho_m^{1/3} \, m_{p'}(1+z),$$ (8)

where $\rho_m$ is the present DM density. If $T_{\text{vir}}/B_{H'} \lesssim 0.1$, essentially no reionization takes place. This dimensionless ratio depends only upon $\alpha'$ and $R$ (apart from the dimensionless environmental parameter $G M_{\text{halo}}^{2/3} \rho_m^{1/3}$). From their Fig. 1, where contours of $T_{\text{vir}} = 0.1 B_{H'}$ in the plane of $m_{e'}$ versus $\alpha'$ are shown for a Milky-Way like galaxy with $m_{p'} = m_p$, one can infer that

$$\alpha' > 1.4 \times 10^{-3} \sqrt{R}$$ (9)

is the condition to avoid reionization during structure formation in our galaxy.

## 2.2 Dark acoustic oscillations

If the dark sector is not too cold ($\xi$ not too small), and if the ionization fraction is not too small, there can be significant pressure waves in the dark sector at the surface of last scattering for the CMB, analogous to baryon acoustic oscillations. These dark acoustic oscillations (DAO) were studied in Ref. [25], assuming that $\xi = 0.5$, and allowing for the possibility that dark atoms only constitute a fraction $f_{int}$ of the total DM. Under these assumptions (for $m_{H'} = 1\,\text{GeV}$), DAO rules out all models in the remaining parameter space of $\alpha'$ versus $R$ if $f_{\text{int}}$ is as large as 0.05. The constraints rapidly weaken as $\xi$ or $f_{int}$ is decreased. These results are illustrated in Fig. 2.

---

[1]Ref. [23] argues that the ellipticity bound is still more than two orders of magnitude stronger than Eq. (7).

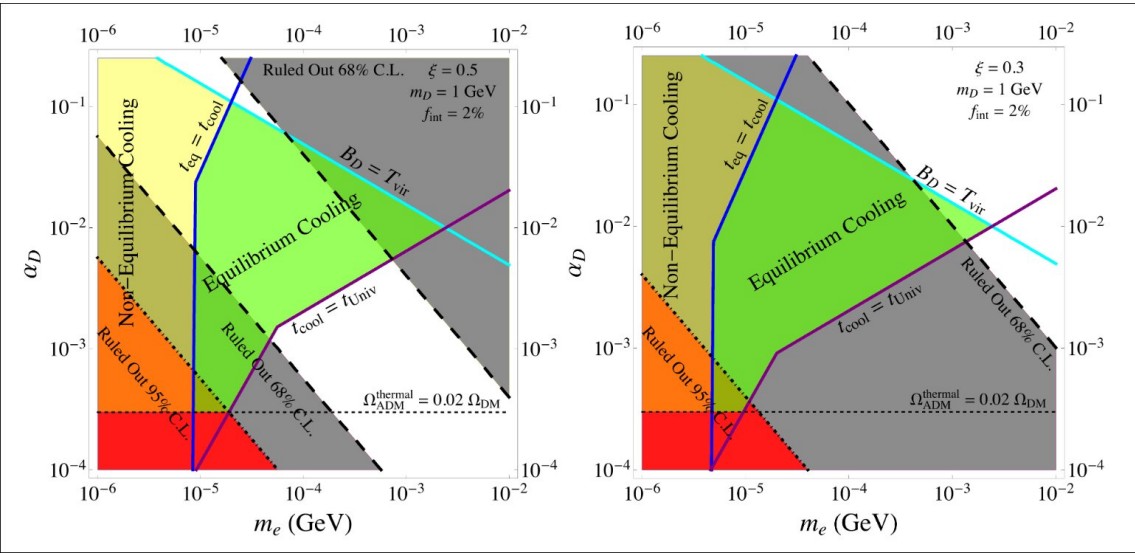

Figure 2: Constraints from DAO on dark atoms with $m_{H'} = 1\,\text{GeV}$ assuming $\xi = 0.5$ (left) or 0.3 (right) and $f_{int} = 0.02$. Allowed regions are unshaded, and $\alpha_D = \alpha'$. Taken from Ref. [25].

## 2.3 Dark atom self-interactions

Although the Bullet Cluster puts an upper limit on the strength of DM self-interactions, it is also known that nearly saturating the bound by taking [26]

$$\frac{\sigma}{m} \sim 0.5\,\frac{\text{cm}^2}{\text{g}} \sim 0.9\,\frac{\text{b}}{\text{GeV}} \tag{10}$$

can have beneficial effects for ameliorating small-scale structure problems of standard cold dark matter. These include the cusp-core, missing satellites and too-big-to-fail problems [27]. Following that review article, the missing satellites seem to have been found [28], or the discrepancy may be a statistical fluctuation [29]. It is also possible that more realistic treatments of structure formation including the effects of baryonic feedback can resolve some of these problems without the need for DM self-interactions [30]. However Ref. [31] notes that tuning the baryonic feedback to solve the cusp-core problem results in discrepancies with the properties of high surface brightness galaxies, and argues that self-interating DM still provides a better fit halo profiles of diverse systems.

For an elementary DM particle of mass $m \sim 1\,\text{GeV}$, Eq. (10) is a very large cross section, but with composite particles it is quite easy to achieve. For dark atoms we can expect a geometric cross section governed by the dark Bohr radius, $a_0' = 1/(\alpha' \mu_{H'})$, with $\sigma \sim \pi a_0'^2$. In fact comparing to measured H atom scattering, this is a significant underestimate: $\sigma/a_0^2 \sim 200-300$ at energies above the atomic unit $E_0 = \alpha^2 \mu_H = 2B_H$ (the Rydberg). Moreover, $\sigma$ has complicated behavior as a function of energy, with numerous resonances, as shown in Fig. 3 (left). Some of these irregularities get smoothed out by considering the transport cross-section $\sigma_t$ instead of the elastic cross section $\sigma$, defined as

$$\sigma_t = \int d\cos\theta\,(1-\cos\theta)\frac{d\sigma}{d\cos\theta} \quad \text{or} \quad \sigma_t' = \int d\cos\theta\,(1-\cos^2\theta)\frac{d\sigma}{d\cos\theta}\,. \tag{11}$$

This weights the cross section by the momentum transfer, which is the physically relevant quantity since purely forward scattering does not have any effect on DM structure formation.

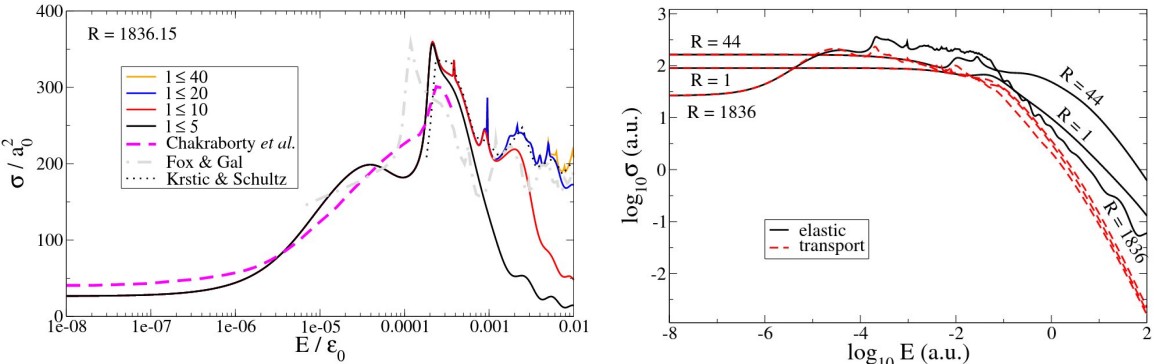

Figure 3: Left: Elastic scattering cross section of H atoms versus energy, in atomic units, from Ref. [17]. The convergence of the partial wave expansion is illustrated by summing different numbers of partial waves. Right: elastic and and transport cross sections versus energy, for several values of $R = m_p/m_e$.

For atom-atom scattering $\sigma'_t$ is the appropriate choice, since exactly backward scattering is the same as forward scattering for identical particles.

It is amusing that, despite the complicated dependences on parameters, these cross sections can be numerically computed for dark atoms for any values of $\alpha'$, $m_{e'}$, $m_{p'}$, by working in atomic units and using results from the atomic physics literature for the scattering potentials, which are already determined in atomic units anyway. Then all dependences scale out of the problem,[2] except for $R$, in the approximation that $B_{H'} \ll m_{H'}$. One has to numerically solve the Schrödinger equation

$$\left[\partial_r^2 - \frac{\ell(\ell+1)}{r^2} - f(R,\alpha')(V_{s,t} - E)\right]u_\ell^{s,t} = 0,\qquad(12)$$

where $u_\ell = r\psi_\ell$, for the partial waves in the spin singlet and triplet channels $(s,t)$, and sum over the orbital angular momentum $\ell$ and spins. Here $f(R,\alpha') = R + 2 + R^{-1} - \alpha'^2/2$, and usually one can neglect the $\alpha'^2/2$ correction. The same technique can be used for scattering of $H_2$ molecules. Ref. [17] found that the energy-dependence can be adequately described by $\sigma'_t \cong (a_0 + a_1 E + a_2 E^2)^{-1}$ with $R$-dependent coefficients.

Interestingly, the reduced cross section at higher energies is compatible with observations that galactic clusters, whose velocity dispersion is higher than dwarf spheroidal or Milky Way-like galaxies, are also more cuspy and thus require a smaller self-interaction cross section. This was noted in Ref. [33] and studied in detail in Ref. [32], which also took into account the inelastic scatterings involving hyperfine transitions, whose energy is

$$E_{hf} = \tfrac{8}{3}\alpha'^4 \frac{m_{e'}^2 m_{p'}^2}{m_{H'}^3} \cong \tfrac{8}{3}\alpha'^2 E_0 \frac{R}{(1+R)^2}.\qquad(13)$$

In Ref. [32] the parameter $R$ was traded for $E_{hf}$, and it was shown that a good overlap between clusters and lower-mass halos could be achieved if $E_{hf}/E_0 \cong 10^{-4}$, implying $\alpha'^2/R \cong 4 \times 10^{-5}$ for $R \gg 1$. This is illustrated in Fig. 4.

---

[2]in the regime $R \gg 1$ where the Born-Oppenheimer approximation works. The static H-H potential is computed assuming that the protons are immobile on the time scale for the electron clouds to readjust themselves at a fixed proton-proton separation.

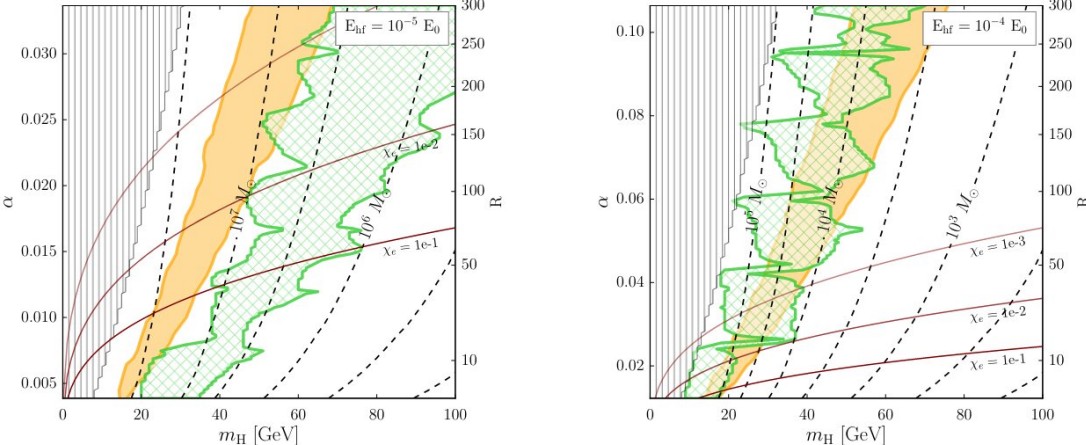

Figure 4: Regions of $\alpha'$ versus $m_{H'}$ where $\sigma'_t$ is compatible with cluster halo profiles (orange) and lower mass halos (green), for $E_{hf}/E_0 = 10^{-4}$ (left) and $10^{-5}$ (right), taken from Ref. [32]. Contours of ionization fraction (here called $\chi_e$) are shown, as well as contours of the minimum halo masses that can form, due to DAO and the dark analog of Silk damping, assuming a dark temperature ratio of $\xi = 0.6$.

## 2.4 Relic density

It is possible that dark atom constituents have equal and opposite aymmetries, consistent with the universe having vanishing net U(1)$'$ charge. Refs. [13, 34] proposed UV completions in which the dark atom asymmetry was directly linked to the baryon asymmetry through leptogenesis. One may ask whether it is possible to achieve the right relic density without any asymmetric component, by the usual thermal freezeout via annihilation into two photons, whose cross section is

$$\langle \sigma v \rangle_{\text{ann}} = \frac{\pi \alpha'^2}{m_{e',p'}^2} S \tag{14}$$

respectively for the $e'$ and $p'$ components. Here $S$ is a Sommerfeld enhancement factor that is typically unimportant ($S \cong 1$ unless $m_i \gtrsim$ TeV [23]).

Unless $R = 1$ ($m_{e'} = m_{p'}$), there will be more unannihilated $p'$s left over than $e'$s. Hence it is natural to focus on the special case $R = 1$ if atomic dark matter is symmetric. Ref. [23] computed the relic density in a model with only one constituent, which we could identify as $p'$, and found the relationship between $\alpha'$ and $m_{p'}$ similar to the black curve in Fig. 5 to match the observed DM density. With two species of the same mass, this curve is adjusted for the fact that $m_{H'} = 2m_{p'}$. In this scenario the DM remains fully ionized unless $\xi$ is very small. On fig. 5 I have overlaid the contours of $f_i$ from Eq. (5) for $\xi = 0.5$. One would need $\xi$ to be smaller to comply with the DAO constraints mentioned above, whereas $\alpha'$ is small enough to satisfy Bullet Cluster and halo ellipticity constraints. In any case, the DM in this model would not be in the form of atoms, but rather ions. It thus seems difficult to explain the relic density of dark atoms without an asymmetry, unless $\xi$ is sufficiently small.

A related question is, given that dark atoms are a form of asymmetric dark matter, how large of an unannihilated symmetric component can be left over? This question is answered for general asymmetric DM models in the seminal reference [35] (see their Fig. 4). For example, if the relic density is mainly provided by the asymmetric component, and the annihilation cross section is only 2.25 times greater than the value needed for symmetric DM, then the symmetric component is suppressed by a factor of 100. This is illustrated by the dashed line in Fig. 5.

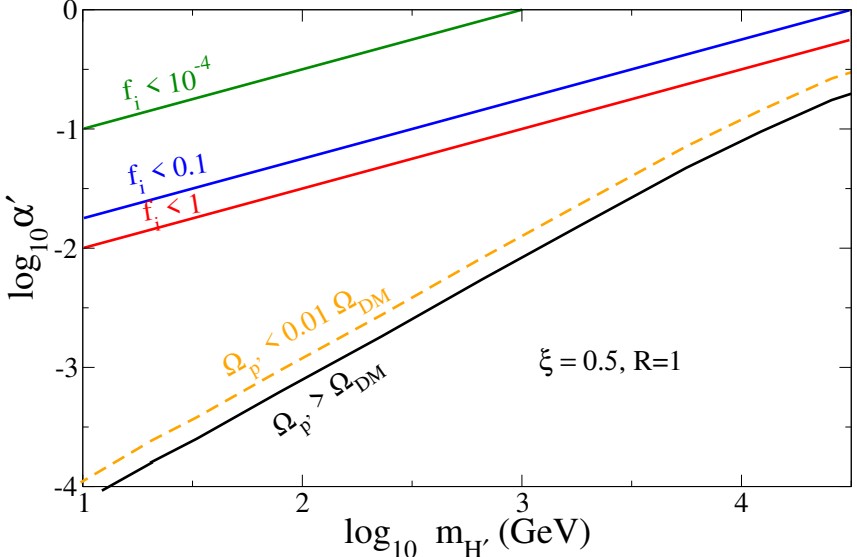

Figure 5: Black curve: contour of correct relic density for a model with only $p'$ symmetric dark matter, reconstructed from Ref. [23]. Dashed curve shows the suppression of the symmetric component when the DM is assumed to be asymmetric. Colored curves are contours of the ionized fraction $f_i$.

## 2.5 Effect of dark photon mass

A common elaboration of the model is to allow the dark photon to have a small mass $m_{\gamma'}$. It could come from a dark Higgs boson that is heavy enough to integrate out from our effective description, or it could come from the Stückelberg mechanism through the interaction

$$\tfrac{1}{2}m_{\gamma'}^2\left(A'_\mu - \partial_\mu\theta\right)^2 , \tag{15}$$

where $\theta \to \theta + \phi'$ under a U(1)$'$ gauge transformation $A'_\mu \to A'_\mu + \partial_\mu\phi'$, so that the gauge symmetry is maintained. The dark Coulomb potential becomes a Yukawa potential with a finite range $\lambda = 1/m_{\gamma'}$. If this is still long compared to the Bohr radius $a'_0 = (\alpha'\mu_{H'})^{-1}$, then the binding properties of dark atoms will be slightly perturbed. One can quantify this effect by approximately solving the Schrödinger equation for the bound state,

$$\left(-\frac{1}{2\mu_{H'}}\partial_r^2 - \alpha'\frac{e^{-r/\lambda}}{r}\right)u = Eu, \tag{16}$$

where again $u = r\psi$. If $\lambda \gg a_0$, we can expand $e^{-r/\lambda} \cong 1 - r/\lambda$ and treat the extra term as a perturbation. The shift in the binding energy is

$$\Delta B_{H'} = -\left\langle\frac{\alpha'}{r}\frac{r}{\lambda}\right\rangle = -\alpha'm_{\gamma'} \tag{17}$$

so one can estimate that dark atoms continue to exist as long as $m_{\gamma'} \lesssim \alpha'\mu_{H'}/2$.

More quantitatively, Ref. [36] solved Eq. (16) numerically and their results for $B_{H'}$ can be fit by the formula

$$B_{H'} \cong \left(1 - 0.85\frac{m_{\gamma'}}{\alpha'\mu_{H'}}\right)^{2.16}\frac{\mu_{H'}\alpha'^2}{2}, \tag{18}$$

which roughly agrees with Eq. (17) for small $m_{\gamma'}$.[3] This indicates that the more accurate

---

[3]They should agree exactly at small $m_{\gamma'}$ since the perturbative calculation is reliable; the discrepancy could be due to digitization inaccuracies since Ref. [36] plots $B_{H'}^{1/2}$ with respect to $1/m_{\gamma'}$ rather than $m_{\gamma'}$.

constraint for having bound states is

$$m_{\gamma'} \lesssim 1.2 \, \alpha' \mu_{H'} \,. \tag{19}$$

A nonnegligible $m_{\gamma'}$ affects the ionization fraction. This was studied in Ref. [37], which noted that the recombination interaction $e' + p' \rightarrow H' + \gamma'$ can be kinematically blocked, leading to higher residual $f_i \sim 0.1$ when $m_{\gamma'} \sim B_{H'}$. Notice that this is still a small mass (by a factor of $\alpha'$) compared to the constraint (19).

## 2.6 Kinetic mixing

So far we have not considered any portals between the dark and visible sectors. The most natural one for dark atoms is gauge kinetic mixing,

$$\tfrac{1}{2} \epsilon F'_{\mu\nu} F^{\mu\nu} \,, \tag{20}$$

where $F'_{\mu\nu}$ is the U(1)$'$ field strength, and $F^{\mu\nu}$ is that of the SM hypercharge, or U(1)$_{EM}$ in an effective Lagrangian description. To diagonalize the gauge boson kinetic terms for $\epsilon \ll 1$, we should distinguish between the two cases $m_{\gamma'} = 0$ or $m_{\gamma'} > 0$ [38]. In the latter case, the field transformation that accomplishes this is

$$A_\mu \rightarrow A_\mu - \epsilon A'_\mu, \qquad A'_\mu \rightarrow A'_\mu \tag{21}$$

so that SM particles acquire small couplings to the dark photon, for example

$$\delta \mathcal{L} = \epsilon e A'_\mu (\bar{p} \gamma^\mu p - \bar{e} \gamma^\mu e) \,. \tag{22}$$

This allows decays $\gamma' \rightarrow e^+ e^-$ if $m_{\gamma'} > 2m_e$. For lighter $\gamma'$, there is the decay $\gamma' \rightarrow 3\gamma$ through an electron loop, with rate [39]

$$\Gamma \cong 1 \, \mathrm{s}^{-1} \left( \frac{\epsilon}{0.003} \right)^2 \left( \frac{m_{\gamma'}}{m_e} \right)^9 \,. \tag{23}$$

If $m_{\gamma'} = 0$ then one has the freedom to choose arbitrary orthogonal linear combinations of $A_\mu$ and $A'_\mu$ as the mass eigenstates. The most convenient choice is through the transformation

$$A_\mu \rightarrow A_\mu, \qquad A'_\mu \rightarrow A'_\mu + \epsilon A_\mu \,, \tag{24}$$

which results in millicharges $q = \epsilon g'/e$ for the dark constituents [40],

$$\delta \mathcal{L} = \epsilon g' A_\mu (\bar{p}' \gamma^\mu p' - \bar{e}' \gamma^\mu e') \,, \tag{25}$$

while the dark photon continues to couple only to the dark constituents.

Another option is Stückelberg mixing [41,42], which uses the same Lagrangian (15) but assumes that under the combined SM U(1) and dark U(1)$'$ gauge transformations $A_\mu \rightarrow A_\mu + \partial_\mu \phi$ and $A'_\mu \rightarrow A'_\mu + \partial_\mu \phi'$ the Stückelberg field transforms as $\theta \rightarrow \theta + \phi' + \lambda \phi$. The kinetic mixing term (20) can also be present. Then after diagonalization of the kinetic terms, the perturbed interaction Lagrangian is [37]

$$\delta \mathcal{L} = \lambda g' A_\mu (\bar{p}' \gamma^\mu p' - \bar{e}' \gamma^\mu e') - (\epsilon - \lambda) e A'_\mu (\bar{p} \gamma^\mu p - \bar{e} \gamma^\mu e) \,. \tag{26}$$

Notice that $m_{\gamma'}$ is assumed to be nonzero in this case.

In general one could have scattering between dark H$'$ and visible protons in direct detection experiments, mediated by both $\gamma$ and $\gamma'$. These interactions will be suppressed by the charge neutrality of H$'$, but do not in general vanish because the charge distributions of $p'$ and $e'$ do

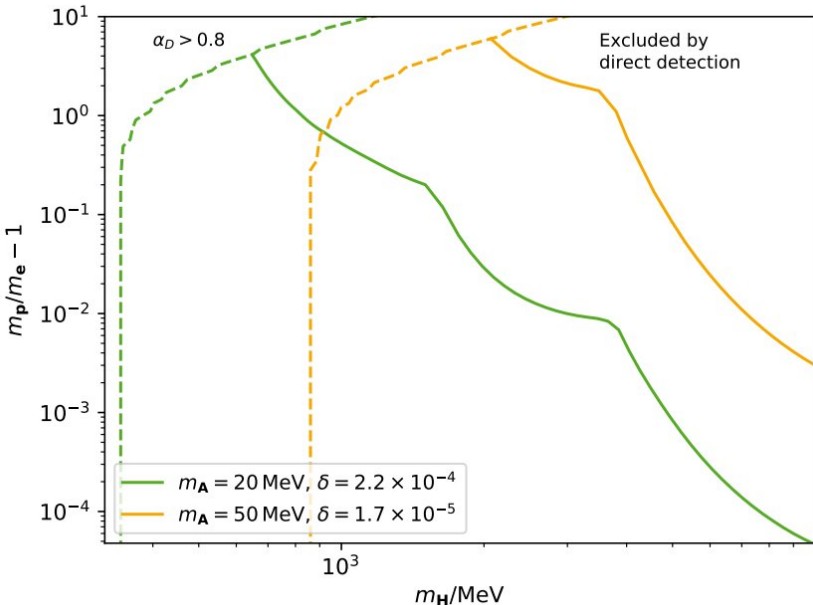

Figure 6: Constraints on $R - 1$ versus $m_{H'}$ from direct detection of dark atoms in a model with Stückelberg mixing and fraction $f_i = 0.1$ of ionized millicharged dark constituents, from Ref. [37]. The parameter $\delta$ is defined to be $\delta = \epsilon - \lambda$.

not exactly coincide, except in the special case $R = 1$. The Fourier transform of the charge distribution becomes a form factor in the matrix element for scattering, that depends on the momentum $q$ transferred. In the limit $q = 0$, the photon (or dark photon) would be sensitive to only the net charge of H$'$, which vanishes. For small $q$, the matrix element is suppressed by $q^2$ and one finds that this cancels the $1/q^2$ photon propagator to give a contact interaction in the case of massless $\gamma'$ [40]. The resulting cross section for $p$-H$'$ scattering when $R \gg 1$ is

$$\sigma_p = 4\pi \alpha'^2 \epsilon^2 \left( \frac{m_p m_{H'}}{m_p + m_{H'}} \right)^2 a_0'^4 \,. \tag{27}$$

If $R = 1$, there is a different (and weaker) velocity-suppressed contribution to direct detection from hyperfine transitions, of order

$$\sigma_p \sim 16 \alpha'^2 \epsilon^2 \left( \frac{m_p}{m_p + m_{H'}} \right)^2 \frac{v^2}{q^2} \sim 16 \frac{\alpha'^2 \epsilon^2}{m_{H'}^2} \left( \frac{m_p}{m_p + m_{H'}} \right)^2 \,. \tag{28}$$

This led to a weak limit $\epsilon g'/e \lesssim 10^{-2}$ in 2012, which at that time was compatible with hints of direct detection by the CoGeNT experiment [43] for $m_{H'} \sim 6$ GeV; the limit is significantly stronger now.

In principle, the ionized components $p'$ and $e'$ could lead to much stronger limits unless $f_i \ll 1$, since there is no cancellation between charges when they scatter on $p$, but this might not be the case if they are millicharged. It was shown that supernovae shock waves expel such particles from the galactic disk [44,45], making them invisible to direct searches. These studies however did not take into account the possible effects of dark photon-mediated interactions between the millicharged particles, which were shown to efficiently randomize their directions in Ref. [46] (see also Ref. [47]). In this case, ionized millicharged particles would not be expelled from the galaxy by supernovae, unless the dark photon were sufficiently massive to damp the self-interactions.

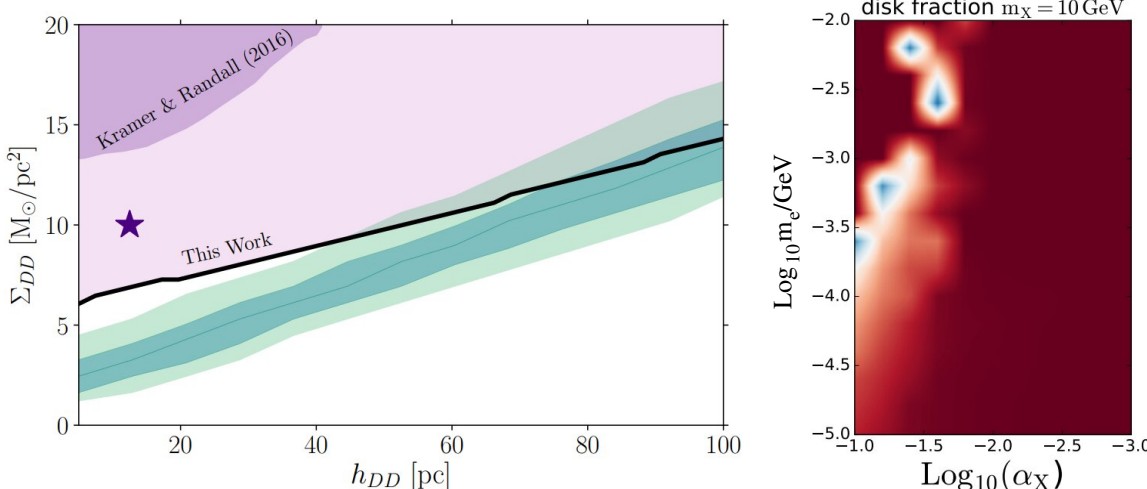

Figure 7: Left: constraints on a dark disk surface density $\Sigma_{DD}$ versus its thickness $h_{DD}$ from Ref. [49]. The star denotes parameters preferred by Ref. [50] for explaining possibly enhanced periodic comet impacts on Earth. Right: regions (blue/white) of $m_{e'}$ versus $\alpha'$ where a dark disk forms, for $m_{H'} = 10\,\mathrm{GeV}$, from Ref. [24].

In the case of Stückelberg mixing, it is possible to have millicharged constituents simultaneously with nonvanishing $m_{\gamma'}$. Inspired by the EDGES 21 cm anomaly [48], Ref. [37] constructed a model with enhanced ionization fraction $f_i \sim 0.1$ by taking $m_{\gamma'} \sim 20-50\,\mathrm{MeV}$, with a view toward naturally explaining a subdominant component of millicharged DM through the ionized fraction of atomic DM, without having to introduce it separately. Achieving $f_i = 0.1$ fixes $\alpha'$ in terms of the other model parameters, and leads to constraints from direct detection in the $R\text{-}m_{H'}$ plane shown in Fig. 6. The effect of hyperfine transitions has been neglected in deriving these constraints.

## 2.7 Further applications

We have already discussed a number of observable effects of dark atoms: DAO, self-interactions, direct detection. The framework is very rich in possible phenomenological consequences. Here we briefly describe several more.

### 2.7.1 Dark disks

In standard CDM structure formation, the DM halo is spheroidal and only visible matter collapses to form the disk of a spiral galaxy. However if some fraction of DM has dissipative interactions similar to baryons, one might expect it to collapse and form a disk that overlaps with the visible one. This idea was explored in depth in Refs. [16,51,52]. In Ref. [16] it was argued that up to $\sim 10\%$ of DM could have strong self-interactions while remaining consistent with Bullet Cluster bounds, and that it could constitute up to 5% of the mass in the Galactic disk. The formation mechanism is similar to that of the visible disk: dark atoms fall toward the galactic center, virialize to temperatures $T > B_{H'}$ through their dissipative interactions, becoming ionized and then cooling via Brehmsstrahlung and Compton scattering on dark photons, that are assumed to be present at the level $\xi \sim 0.5$.

This proposal has come under pressure from Gaia measurements, that are able to constrain the surface density (mass per unit area) $\Sigma_{DD}$ of the dark disk [49,53–55]. Constraints on $\Sigma_{DD}$ depend on its assumed thickness $h_{DD}$, as illustrated in Fig. 7 (left). Ref. [54] obtained stronger

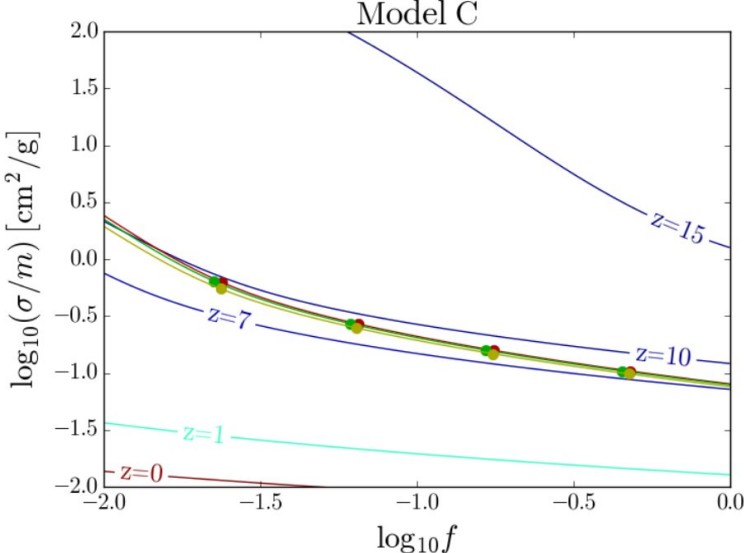

Figure 8: Trajectories in the plane of $\sigma/m$ versus fraction of atomic DM $f$, with scattering assumed to be dissipative, showing the degeneracy of DM parameters with the amount of accretion subsequent to BH formation, for models consistent with redshifts of three observed SMBHs (taken from Ref. [56]). Successive clusters represents steps of 1 $e$-folding in mass growth, from right to left, and contours of constant redshift $z$ are shown. The BHs corresponding to the rightmost cluster must have undergone 1 or 2 $e$-foldings of growth to match the observations.

limits, similar to the green curves in Fig. 7 (left).

The previous works assume that a dark disk forms, but this need not be the case. Ref. [24] shows that with a subdominant component (5%) of atomic dark matter, cooling occurs too early for dark disks to survive; they tend to be transformed into bulges by subsequent gravitational torques from dense DM clumps. The small (blue) parameter regions where disks form are shown in Fig. 7 (right) for $m_{H'} = 10$ GeV; these regions enlarge and merge to some extent for lighter $m_{H'} = 1$ GeV atoms.

A thorough study of dark atomic structure formation within our galaxy was made in Ref. [47], in the context of a Twin Higgs mirror sector that constitutes a fraction of the total dark matter. It includes kinetic mixing that induces nanocharges for the dark constituents and enables direct detection. Depending upon details of the dark astrophysics, the mirror constituents may form a disk or remain in a halo, and they may be in ionized or atomic form.

### 2.7.2 Early SMBH formation

Observations of supermassive black holes (SMBHs) at surprisingly high redshifts [57,58] have sparked interest in the possibility that a fraction $f$ of strongly interacting dark matter could catalyze their formation [59]. The very large cross sections are naturally accommodated by dark atoms constituting this part of the total DM [60]. The dissipative interactions of atomic DM can accelerate the process of gravothermal collapse that would initiate formation of a black hole, at an earlier time than in standard CDM cosmology. Subsequent accretion could then allow the BH to reach its observed mass by redshifts $z \sim 7$. This scenario was confirmed using a modified $N$-body gravitational simulation in Ref. [56], which found that dissipative scattering is much more effective than elastic scattering for seeding SMBHs, and could allow for a fraction as large as $f = 1$ of atomic DM while marginally satisfying Bullet Cluster constraints. Fig. 8

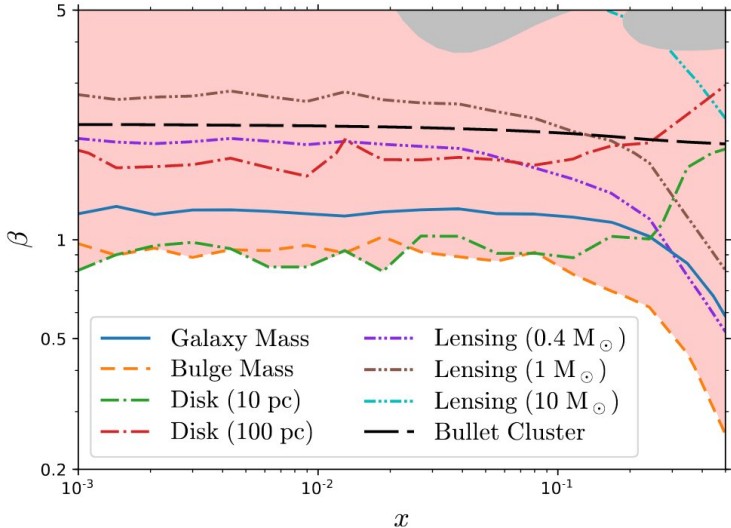

Figure 9: Constraints on the ratio $\beta$ of mirror baryons to visible baryons versus $x = \xi = T'/T$ from structure formation in models with perfect mirror symmetry, from Ref. [64]. $\beta$ is related to $f$, the fraction of atomic DM, by $f = \beta\,\Omega_b/\Omega_{DM} \cong 0.18\,\beta$.

shows the allowed parameters assuming dissipative scattering. For smaller $f$, a somewhat larger $\sigma/m$ and number of $e$-foldings of accretion would be needed, as can be estimated from the curve by counting clusters.

### 2.7.3 3.5 keV X-ray line

The origin of a 3.5 keV X-ray signal [61,62] in XMM-Newton observations of galactic clusters and M31 remains controversial, but decays of 7 keV sterile neutrino DM into photons have been a highly studied candidate. In Ref. [63] we considered the alternative possibility that 3.5 keV corresponds to the dark hyperfine transition energy (13), if $m_{\gamma'} > E_{hf}$ to kinematically block the decay of the triplet excited state $H'_3 \to H'_1 + \gamma'$ into dark photons, while introducing kinetic mixing to allow the visible decay $H'_3 \to H'_1 + \gamma$. The excited state could either be primordial, with a lifetime similar to the age of the universe, or it could be short-lived and result from late-time self-interactions of $H'$. The latter scenario requires relatively large kinetic mixing, and is constrained by direct detection toward heavy dark atoms, $m_{H'} > 350\,\text{GeV}$ (in 2014, no doubt larger now in light of stronger direct limits).

### 2.7.4 Dark molecules, planets, stars . . .

Apart from dark disks, other more complex structures beyond atoms can form, depending upon the parameters in the dark sector, including the important environmental ones $\xi = T'/T$ and $f$, the fraction of DM comprised by dark atoms, versus conventional CDM. In mirror models or other variants having nontrivial chemistry, the abundance of He$'$ plays an important role in structure formation. Ref. [24] studied structure formation in a simple dark sector without chemistry, taking $f = 0.05$ (the limit from DAO assuming $\xi = 0.5$) and $R \gg 1$, finding that much of the parameter space is ruled out by the formation of MACHO-like structures or a dark bulge in excess of constraints on the observed mass-to-light ratio of the luminous part of the galaxy.

Ref. [64] repeated this analysis for the model of exact mirror symmetry, with $\xi$ and $f$ being the only free parameters. Largely due to the effects of He$'$, not present in simple atomic DM

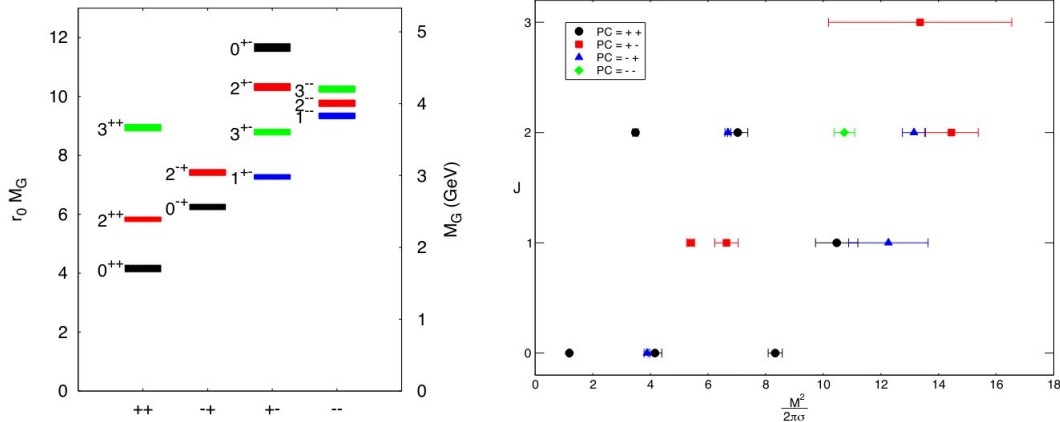

Figure 10: Spectrum of glueballs for pure SU(3) [67] (left) as a function of $J^{PC}$ quantum numbers and for large $N$ [68] showing $J$ versus $m^2$ (right). The parameters $r_0 \sim 1/\Lambda'$ and $\sigma \sim \Lambda'^2$. Figure from Ref. [69].

models, the structure formation constraints found in the latter are significantly relaxed, and allow for $f \lesssim 0.14$, as shown in Fig. 9. He$'$ ions are efficient for capturing free electrons, lowering the ionization fraction, and impeding the formation of H$_2'$ molecules, which are important building blocks for structure. This ultimately reduces the number of dense dark structures that are constrained by MACHO searches or mass-to-light observations.

The previous studies were done using the extended Press-Schechter formalism for simulating the merger history of DM halos. Eventually it may be interesting to repeat these using gravitational $N$-body simulations including hydrodynamics, that could provide a closer to first-principles analysis. Cooling rates in these complex dark sectors including molecules with dissipative interactions have been computed in Refs. [65,66] as a necessary first step to enable such simulations.

## 3 Dark Glueballs

The simplest nonabelian confining sector is one consisting of gauge bosons alone. At temperatures below the confinement scale $\Lambda'$, there is a lightest glueball state with quantum numbers $0^{++}$, and a spectrum of excited states, that have been studied on the lattice for SU(3) and SU($N$) gauge theories [69,70]. For example in real-world QCD, but with quarks omitted, the lightest glueball mass is predicted to be 1750 MeV [71]. Taking $\Lambda_{QCD} = 260$ MeV [72], we could expect the lightest glueball mass to scale as $m_0 = 6.7 \Lambda'$ for an SU(3) hidden sector with a different confinement scale. Glueball spectra for pure SU(3) and large-$N$ SU($N$) as determined by lattice gauge theory are shown in Fig. 10. The first proposal of hidden sector glueballs as DM was as early as Ref. [73], motivated by hints of DM self-interactions for cosmological structure formation and by string theory.

One can quickly be convinced that it is not possible to explain the relic density of dark glueballs using conventional thermal freezeout, even if there is some portal to the SM such as $M^{-1}\phi^2 \bar{f}f$, where $\phi$ is the effective glueball field, $f$ is a SM fermion and $M$ is a mass scale. Since $\phi$ carries no global charge, it cannot be stabilized against decay and the existence of such an operator would imply that $\phi \bar{f}f$ is also present, and generically more important. Thus glueballs with portal interactions will be unstable, and if their lifetime is longer than the age of the universe, their annihilation rate will be even slower. This is borne out in real QCD, where

there is no stable glueball because it mixes with mesons of the same quantum numbers. The same argument implies that elastic scattering rates of glueballs on visible baryons for direct detection are negligible [33].

## 3.1 Relic density

Elaborating on the previous statements, suppose there is an effective coupling

$$\mathcal{O} = \frac{1}{M^3} G_{\mu\nu} G^{\mu\nu} \bar{f} f \tag{29}$$

between the SU(N)$'$ field strength and a SM fermion, for example. Then by dimensional analysis we have matrix elements for decay and scattering of order

$$\langle f\bar{f}|\mathcal{O}|\phi\rangle \sim \frac{\Lambda'^4}{M^3}, \quad \langle f\bar{f}|\mathcal{O}|\phi\phi\rangle \sim \frac{\Lambda'^3}{M^3}, \tag{30}$$

leading to decay rate and scattering cross section

$$\Gamma \sim \frac{\Lambda'^7}{M^6}, \quad \langle \sigma v \rangle \sim \frac{\Lambda'^4}{M^6}. \tag{31}$$

Equating $\langle \sigma v \rangle$ to the canonical cross section for thermal freezeout $\sim 3 \times 10^{-26}\,\text{cm}^3/\text{s}$ gives $\Lambda'^2 \sim 5 \times 10^{-5} M^3/\text{GeV}$, while demanding that $1/\Gamma$ exceed the age of the universe requires $\Lambda'^7 \lesssim 10^{-42} M^6\,\text{GeV}$. The nontrivial solution of these equations is $M \sim 1\,\text{keV}$, $\Lambda' \sim 0.01\,\text{eV}$, which is too small for thermal freezeout.

If there is no portal to the SM, then glueballs will form from gluons when the dark sector temperature $T'$ falls below $\Lambda'$.[4] Their initial density can be rougly estimated by equating the energy density of gluons to that of glueballs at the time of the confinement phase transition. Ref. [74] first pointed out that $3 \to 2$ scattering processes mediated by an effective operator $\sim \phi^5/(5!N^3\Lambda)$ would determine the subsequent relic abundance. The $3 \to 2$ process, originally studied for DM evolution in Ref. [75] and dubbed "cannibalism," will come back later in our discussion of dark mesons. A notable feature of the mechanism is that it causes the DM to cool more slowly by the conversion of mass into kinetic energy, if the dark sector is secluded. In the absence of a portal interaction for keeping the DM in kinetic equilibrium with the SM, this may result in warm dark matter, which is now disfavored by Lyman-$\alpha$ constraints [76], as well as Milky Way satellite counts [77] and measurements of the dark-matter subhalo mass function in the inner Milky Way [78]. Ref. [74] finds that the glueballs are cold DM for masses above 1 MeV.

If the hidden sector starts out sufficiently cold, $\xi \ll 1$, the number density of gluons is suppressed and $3 \to 2$ processes may never come into equilibrium. In this case the relic glueball density can be estimated by converting the energy density of gluons at the transition when $T' = \Lambda'$ and $T = T'/\xi$. This estimate was made in Ref. [79], giving

$$\Omega_{gb} \sim 4 \times 10^8 \frac{(N^2-1)}{g_*} \frac{\Lambda'}{\text{GeV}} \xi^3 = 2 \times 10^8 \left(\frac{s'}{s}\right) \frac{\Lambda'}{\text{GeV}}, \tag{32}$$

where $\xi$ and $g_*$ (counting SM degrees of freedom) are evaluated at the transition, and $s'/s = 2(N^2-1)\xi^3/g_*$ is the ratio of entropies in the two sectors. A small value of $\xi$ is thus also needed for getting the desired abundance $\Omega_{gb} = 0.27$, for reasonably large $\Lambda'$. Interestingly, Refs. [80, 81] get a very similar result even taking into account $3 \to 2$ scattering, $\Omega_{bg} = 3 \times 10^8 R \Lambda_x/\text{GeV},$[5] where $R = s'/s$ is the ratio of entropies immediately *after* the phase

---

[4]or if the dark sector has not yet thermalized, then when the density falls below $n' \sim \Lambda'^3$.

[5]from digitizing Fig. 1 of their first paper

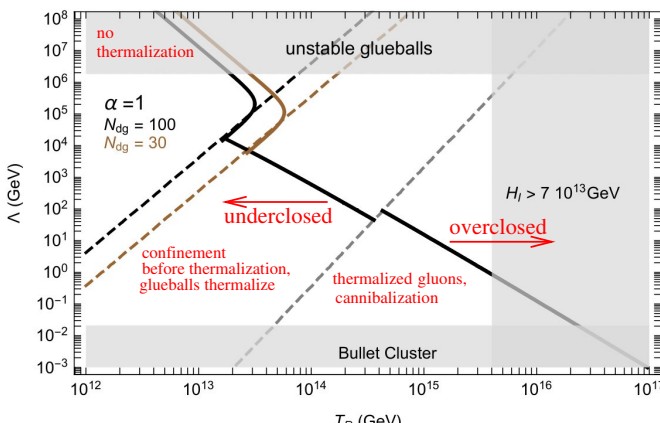
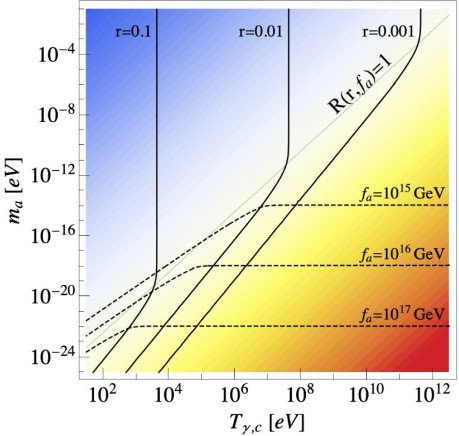

Figure 11: Left: parameter space for gravitational production of dark glueballs, adapted from Ref. [84]. "Unstable glueballs" refers to gravitational decays (which are unavoidable in any model of dark glueballs) faster than the present Hubble rate. $\alpha$ refers to the $3 \to 2$ cross section $\langle \sigma v^2 \rangle_{3 \to 2} \equiv \alpha^3 / m_0^3$. $N_{DG}$ is the assumed number of glueballs produced per gluon in the phase transition with unthermalized gluons. Right: regions of glueball- (blue) versus axion-dominated (red) DM in the plane of axion mass versus photon temperature at the time of the confinement phase transition, from Ref. [85]. Here $r = \xi$ and $\xi$, $f_a$ are adjusted to give observed DM abundance at each point in the plane.

transition, and $\Lambda_x$ is identified with the glueball mass $m_0$. The strength of the $3 \to 2$ transition is estimated using large-$N$ [82] and naive dimensional analysis (NDA) arguments [83] giving the glueball potential

$$V(\phi) \sim \sum_{n=2} \frac{1}{n!} \left( \frac{4\pi}{N} \right)^{n-2} \Lambda_x^{4-n} \phi^n . \tag{33}$$

In addition to the lowest mass glueball state $0^{++}$, there are many other stable excited states, whose relic density has been shown to be much smaller in Refs. [80, 81].

Eq. (32) suggests that there would be no production of glueballs in an inflationary scenario where reheating was purely into SM particles. But gravity couples to everything, and Ref. [84] uses results from conformal field theory to show that purely gravitational couplings can be sufficient to produce dark glueballs, depending on the reheat temperature $T_R$. SM particles annihilating into an $s$-channel graviton produce dark gluons with a relative abundance going as $Y' \cong 10^{-6}(N^2-1)(T_R/M_p)^3$, where $M_p$ is the Planck mass. One can also compute the relative energy densities in the two sectors. If $\phi\phi \to \phi\phi$ scattering is fast enough, the dark glueballs will thermalize before the confinement transition, and $3 \to 2$ scattering subsequently comes into equilibrium, lowering the density. However the confinement phase transition could happen before thermalization. Then the typical gluon is more energetic than $\Lambda'$, and the number of glueballs produced per gluon $N_{DG}$ depends upon the details of hadronization, which can affect the final relic density. These outcomes are illustrated in Fig. 11 (left).

Ref. [86] argues that string theory generically predicts not just one hidden sector, but many, which can exacerbate the generic problem that dark glueballs are overproduced unless their sectors are left relatively unpopulated by reheating after inflation. An alternative possibility is that the universe comes to be matter-dominated by moduli and undergoes a second stage of late reheating by their decays, which could be preferentially into SM particles [87]. A similar mechanism using domination by vector-like quarks charged under both SU(N)′ and color SU(3) was studied in Ref. [88].

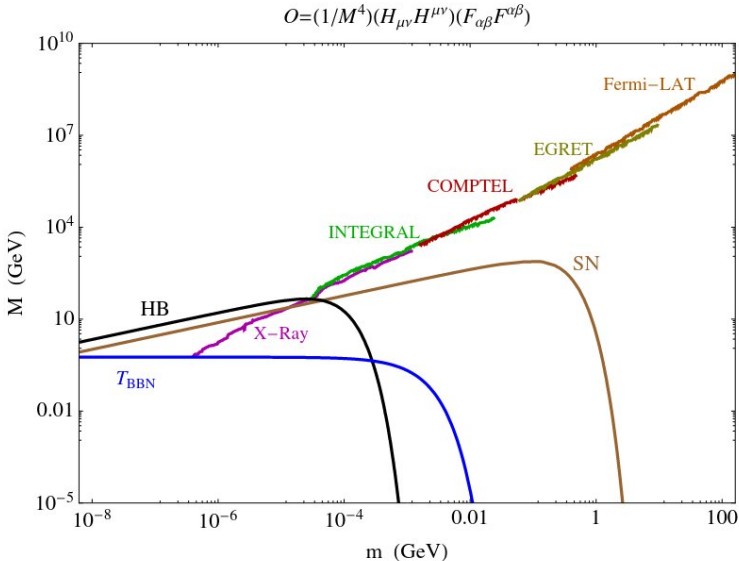

$$O=(1/M^4)(H_{\mu\nu}H^{\mu\nu})(F_{\alpha\beta}F^{\alpha\beta})$$

Figure 12: Constraints on the scale $M$ for the $M^{-4}B^2\,\mathrm{tr}\,G^2$ portal versus glueball mass, from cosmic ray mononergetic line searches and energy loss in horizontal branch stars or type II supernovae [74].

A simple solution to the dark glueball overproduction problem is to include a coupling of $G_{\mu\nu}$ to axions,

$$\frac{g'^2}{32\pi^2 f_a}a\,G_{\mu\nu}\widetilde{G}^{\mu\nu}. \tag{34}$$

The additional interaction allows for a redistribution of abundances to deplete the glueball density in favor of ultralight axions, resulting in a two-component DM scenario [85, 89]. Fig. 11 (right) illustrates the regions of parameter space favoring glueballs or axions constituting most of the DM. The dividing line between glueball versus axion domination is given by the criterion

$$R(\xi, f_a) = \left(\frac{\xi}{10^{-2}}\right)^2\left(\frac{6\times 10^{13}\,\mathrm{GeV}}{f_a}\right) = 1\,, \tag{35}$$

where the temperature ratio $\xi$ is evaluated at time of the confinement transition.

## 3.2 Self-interactions and glueballinos

The glueballs are strongly interacting particles with a geometric cross section of order $\sigma \sim 4\pi/\Lambda'^2$; hence they may be able to address the small-scale structure problems of CDM mentioned previously. Matching to the desired value of $\sigma/m$ and using the relation between $m$ and $\Lambda'$, one finds that $\Lambda' \sim 100\,\mathrm{MeV}$ [33, 79], favoring glueballs below the GeV scale. However these references ignored the scaling with $N$ and factors of $4\pi$ from NDA. Taking the potential (33) at face value, one finds a cross section

$$\sigma \sim \frac{12\pi^3}{N^4 m_0^2}, \tag{36}$$

which would satisfy the criterion (10) if

$$m_0 \sim 130\,\mathrm{MeV}\left(\frac{3}{N}\right)^{4/3}. \tag{37}$$

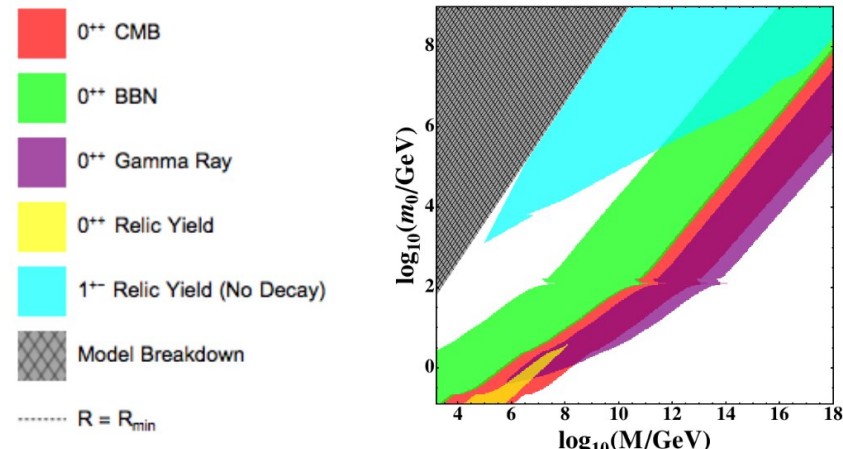

Figure 13: Example of constraints on $0^{++}$ and $1^{+-}$ glueballs from CMB, BBN and relic density, assuming a minimal value of the entropy ratio $R = s'/s$ such that $T' \sim \Lambda'$, conserved dark $C$-parity, and absence of dimension-6 operators in Eq. (38), from Ref. [81]. "Model Breakdown" indicates where $m_0 > M/10$, and the effective field theory treatment of the portal interaction may not be valid.

Refs. [90, 91] have computed the scattering cross section on the lattice for SU(2) glueballs, with large systematic errors, giving $\sigma = (2 - 51)/\Lambda^2$. This is signficantly higher than the prediction (36), which gives $\sigma \cong 0.6/\Lambda^2$.

If one takes seriously the indications that DM self-interactions should be velocity-dependent, glueballs are not the best candidates since they have a contact interaction leading to constant $\sigma$. A simple extension is to include an adjoint fermion $X$ (gluino) which can bind with a gluon to form a a stable color singlet "glueballino" state $\tilde{\phi}$. Ref. [79] considers the case where the fermion mass $m_X \gg \Lambda'$. Then glueballinos are heavier than glueballs, and experience velocity-dependent self-scattering by virtual glueball exchange. Glueballinos can undergo thermal freezeout by $\tilde{\phi}\tilde{\phi} \to \phi\phi$ annihilation.

The $\tilde{\phi}$ DM scenario was also studied in Ref. [92] where it was called "gluequark DM," with emphasis on the fact that there are generally two stages of annihilation: first at the constituent level $XX \to gg$, and again following the confinement transition through $\tilde{\phi}\tilde{\phi} \to \phi\phi$. Moreover if glueballs decay into SM states, this can significantly dilute abundances in the hidden sector. In general one must consider all of these effects to determine the relic density. Ref. [93] showed that the observed relic density can be achieved even for glueballino masses as high as the PeV scale. This is well above the conventional perturbative unitarity constraint for the annihilation cross section [94].

## 3.3 Indirect signals

Various portals connecting glueballs to the SM are possible. If $\mathcal{O}_{SM}$ is a SM gauge singlet operator of dimension $n$, then one can consider the interaction $M^{3-n}\phi\,\mathcal{O}_{SM}$ at the effective field theory level, which mediates glueball decay. Strong constraints on light glueballs arise from the CMB if $\phi$ can decay into charged particles or photons. Observations of cosmic ray photons constrain monoenergetic signals from $\phi \to \gamma\gamma$ [74], as shown in Fig. 12. Moreover shorter-lived glueball excitations, even if unimportant as DM candidates, can disrupt big bang nucleosynthesis (BBN) by injecting energy [80, 81].

Rather than working at scales below the glueball mass, it can be more theoretically informative to think in terms of portals involving the dark nonabelian field strength $G_{\mu\nu}$ and SM

U(1) field strength $B_{\mu\nu}$, Higgs field $H$, or fermions $f$. Ref. [81] finds that the leading operators are

$$\frac{1}{M^4}B^2 \operatorname{tr}(G^2), \quad \frac{1}{M^4}B_{\mu\nu}\operatorname{tr}(G^3)^{\mu\nu}, \quad \frac{1}{M^2}|H|^2\operatorname{tr}(G^2). \tag{38}$$

This study emphasized the relevance of the usually subdominant $1^{+-}$ glueballs, that can be long-lived and even be the dominant DM. (Unlike the $0^{-+}$ excited state, the $1^{+-}$ state is not diluted by coannihilation with the ground state, if $C$ is conserved.) An example of the ensuing constraints on glueball mass versus the scale $M$ in Eq. (38) is shown in Fig. 13. They are sensitive to the initial entropy ratio $R = s'/s$ at the confinement transition and whether dark $C$-parity is conserved, which would forbid the second operator in (38).

In addition to constraints, one can address anomalies like the 3.5 keV line mentioned in section 2.7.3. Ref. [95] noted that glueballinos with the desired properties for the relic abundance and self-interactions can also have a hyperfine transition energy of 3.5 keV. Another interesting example is given by Ref. [85], which found that a subdominant, strongly self-interacting glueball component has the right properties to catalyze early formation of SMBHs, as discussed in Section 2.7.2.

## 4 Dark mesons

If quarks are added to the hidden $SU(N)'$ sector, then dark mesons $\pi'$ become a DM candidate, which can be lighter than the glueballs $\phi$ if the quark mass $m_{q'}$ is below $\Lambda'$. Like visible pions, the dark $\pi'$ would have quantum numbers $0^{-+}$, and a $0^{++}$ glueball could undergo decay as $\phi \to \pi'\pi'$ if $2m_{\pi'} < m_\phi$, leaving $\pi'$ as the sole DM candidate. Even if decays are kinematically blocked, annihilations $\phi\phi \to \pi'\pi'$ can greatly deplete the relic glueball abundance.

With only a single quark flavor and SU(3), one would expect $m_{\pi'} \sim 4\Lambda'$, similar to the $\eta'$ of QCD, which is not light enough to satisfy $2m_{\pi'} < m_\phi$, but sufficient for $m_{\pi'} < m_\phi$. Such a dark pion would, like the glueball, be unstable to decays into gravitons, with amplitude $\mathcal{M}_{\pi' \to gg} \sim m_{\pi'}^3/M_p^2$ and hence lifetime

$$\tau \sim 16\pi \frac{M_p^4}{m_{\pi'}^5} \sim \pi \times 10^{17}\text{s} \left(\frac{2 \times 10^7\,\text{GeV}}{m_{\pi'}}\right)^5, \tag{39}$$

showing that metastability on cosmological timescales imposes a modest requirement on the mass.

With more flavors, absolute stability becomes possible, and the pions could be pseudo-Nambu-Goldstone bosons (pNGBs) from spontaneously broken chiral symmetry. The stability criterion (even with quarks that are also coupled to the SM SU(2)) can be formalized analogously to QCD in terms of $G$-parity: the lightest $G$-odd pion (LGP) is stable [96] for $SU(N)'$ with $N \geq 3$. In the SM this would be the $\pi^0$, in the absence of electromagnetism. It is not gravitationally stable, but if couplings to weak interactions were turned off, then $\pi'^\pm$ would become degenerate with $\pi'^0$, and the former would be completely stable.

In addition to pseudoscalar mesons, there will be heavier vector mesons. In some circumstances they could be the primary DM candidates. Ref. [97] presents a model with $SU(2)'$ and a complex scalar doublet $\phi$ that can have a Higgs portal coupling $\lambda|H|^2|\phi|^2$. This allows the lighter scalar pion $\phi^\dagger\phi$ to decay into SM particles, but leaves the vector stable since a vector cannot mix with the Higgs. Even if the vector mesons are not DM, they can play an important role in the freezeout process, as we will see.

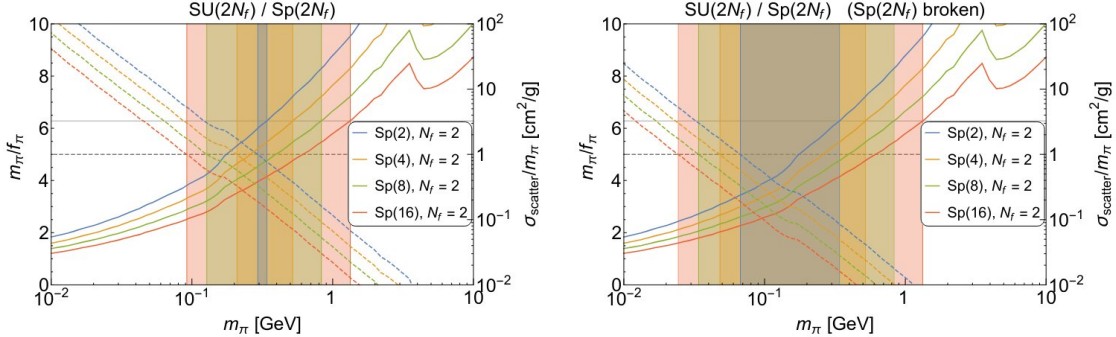

Figure 14: Allowed ranges of mesonic DM produced by the SIMP mechanism, for different confining Sp($N$) gauge groups, from Ref. [98]. Left: unbroken flavor symmetry (degenerate quark masses); right: broken flavor symmetry (lifting degeneracy of $\pi'$ masses).

## 4.1 Relic density

Chiral Lagrangians (see for example Ref. [99]) are the appropriate effective theory for dark mesons that are pNGBs like the SM pseudoscalar octet. They are constructed from the matrix $\Sigma = \exp(i\pi'/f)$, where $\pi' = \pi'^a T^a$ and $T^a$ are the generators of the flavor symmetry that is spontaneously broken by $\langle\Sigma\rangle_{ij} = \delta_{ij}$, which is proportional to the matrix of quark condensates $\langle\bar{q}'^i q'^j\rangle$ in flavor space. Including the symmetry-breaking quark mass matrix $M$, the leading terms in the chiral Lagrangian are

$$f^2 \,\mathrm{tr}(\partial_\mu \Sigma^\dagger \partial^\mu \Sigma) - f^3 \,\mathrm{tr}(M\Sigma + \mathrm{H.c.}), \tag{40}$$

giving the pions a mass $m_{\pi'}^2 = f M$. $f$ is known as the pion decay constant ($f \sim m_\pi$ in the real world): the hadronic matrix element of the axial quark currents can be parametrized as

$$\langle 0|\bar{q}\gamma^\mu \gamma_5 T^a q|\pi'^b\rangle \sim f p^\mu \delta_{ab}. \tag{41}$$

This assumes that chiral symmetry breaks to SU(N); if it breaks to Sp(N) then $\langle\Sigma\rangle_{ij} = J_{ij}$, where $J$ is a symplectic matrix.

### 4.1.1 Thermal $2 \to 2$ freezeout

The early reference [100] considered the portal interactions

$$\lambda_h(|H|^2 - v^2)\mathrm{tr}(\partial_\mu \Sigma^\dagger \partial^\mu \Sigma) \quad + \quad \frac{\lambda_v}{f} B^{\mu\nu}\mathrm{tr}(M\Sigma\partial_\mu \Sigma^\dagger \partial_\nu \Sigma + \mathrm{H.c}) \tag{42}$$

to the Higgs and the hypercharge field strength. The Higgs portal enables annihilations $\pi'\pi' \to WW, HH$ for thermal freezeout. The hypercharge portal would allow for $Z \to \pi'^0 \pi'^+ \pi'^-$, for example, but this is kinematically forbidden if the $2 \to 2$ processes are allowed. Such heavy pions would be incompatible with strongly self-interacting DM (SIDM); see below.

Ref. [33], motivated by SIDM to consider $m_{\pi'} \sim 30$ MeV, suggested annihilation $\pi'\pi' \to Z'Z'$ into light $Z'$ gauge bosons via a $F'_{\mu\nu}F'^{\mu\nu}\mathrm{tr}(\partial\Sigma^\dagger \partial\Sigma)$ interaction. However one needs to keep $Z'$ in thermal equilibrium with the SM for standard freezeout. Using kinetic mixing to allow for $Z' \to e^+e^-$ leads to conflict with CMB constraints because of late $\pi'\pi' \to Z'Z' \to 4e$ annihilations. Taking the $Z'$ to be massless with sufficiently small kinetic mixing can overcome these problems.

Confining SU(2) models are special since the "baryons" are scalars like the mesons, and differ only in terms of which conserved quantum numbers assure their stability. Ref. [101] considered SU(2) with two flavors of quarks, $Q_u$ and $Q_d$, assigned equal and opposite SM hypercharge. The baryons $N = Q_u Q_d$ and $\bar{N} = \overline{Q}_u \overline{Q}_d$ are stable (and neutral) DM candidates, while the mesons $\pi'^0$, $\pi'^{\pm}$ can be made unstable to decays into SM fermions $f$. Then $N\bar{N} \to \pi'\pi'$ along with $\pi' \to f\bar{f}$ can achieve the desired relic density through conventional freezeout.

### 4.1.2 Thermal $3 \to 2$ freezeout

A qualitatively different means of freezeout was proposed in Ref. [98], based on the Wess-Zumino-Witten (WZW) interaction

$$\frac{2N}{15\pi^2 f_{\pi'}^5} \epsilon^{\mu\nu\rho\sigma} \mathrm{tr}(\pi' \partial_\mu \pi' \partial_\nu \pi' \partial_\rho \pi' \partial_\sigma \pi') \tag{43}$$

(where $f_{\pi'} \sim f$ up to factors of 2). It is a topological term, that only exists in theories where the 5th homotopy group $\pi_5(\mathcal{G}/\mathcal{H})$ is nontrivial. These include the gauge groups SU($N$) and SO($N$) if the number of flavors $N_F \geq 3$, and Sp($N$) if $N_f \geq 2$. Notice that $N$ must be even in the case of Sp($N$). Then the $3 \to 2$ process (cannibalization), previously discussed for glueballs, becomes possible. This is an example of the SIMP mechanism introduced in Ref. [102]. This mechanism assumes that the DM is in thermal equilibrium with the SM at the time of freezeout, but how this is accomplished for the dark pion model is not discussed in Ref. [98].

The $3 \to 2$ cross section from the WZW interaction scales as

$$\langle \sigma v^2 \rangle_{3 \to 2} \equiv \frac{\alpha_{\mathrm{eff}}^3}{m_{\pi'}^5} \sim \frac{N^2 N_f^5 m_{\pi'}^5}{f_{\pi'}^{10}}. \tag{44}$$

In place of a dimensionless coupling, the ratio $m_{\pi'}/f_{\pi'}$ determines the strength of the interaction, and chiral perturbation theory breaks down for $m_{\pi'}/f_{\pi'} \gtrsim 2\pi$. This puts an upper limit on $m_{\pi'}$ for which the relic density is small enough. On the other hand the self-interactions (discussed below) put a lower limit on $m_{\pi'}$. This gives rise to somewhat narrow ranges for $m_{\pi'} \sim 30 - 1000\,\mathrm{MeV}$, depending on the numbers of colors and flavors, and also on whether the flavor symmetry is exact or broken. These ranges are illustrated in Fig. 14.

### 4.1.3 Portals for thermal equilibration

To study the effect of a mediator to maintain kinetic equilibrium between the $\pi'$ and SM sectors in the SIMP scenario, Ref. [103] charged the quarks under a dark U(1)$'$, assuming $N_f = 3$ flavors and charge matrix $Q = \mathrm{diag}(1, -1, -1)$, chosen to cancel mixed anomalies of the AVV type between the global axial and vector flavor currents. This suppresses the decay of the $\pi^0$- and $\eta^0$-like mesons into $Z'Z'$.[6] Kinetic mixing of the $Z'$ with the SM hypercharge can keep the two sectors in equilibrium. The interactions of $Z'$ with the pions is obtained from chiral perturbation theory by covariantizing the derivatives, $\partial_\mu \Sigma \to \partial_\mu \Sigma + ig'[Q, \Sigma]Z'_\mu$, yielding standard U(1) couplings to the charged pion currents and seagull terms. Ref. [103] analytically estimated the $\pi'$ abundance from $3 \to 2$ freezeout, obtaining the observed value for pion masses

$$m_{\pi'} = 0.03\,\alpha_{\mathrm{eff}}(T_{eq}^2 M_P)^{1/3} \sim 35 - 350\,\mathrm{MeV}, \tag{45}$$

where $T_{eq} = 0.8\,\mathrm{eV}$ is the matter-radiation equality temperature, and the range of $m_{\pi'}$ is from taking $\alpha_{\mathrm{eff}} = 1 - 10$. The $Z'$ is taken to be heavier than $m_{\pi'}$ so that $\pi'\pi' \to SM$ annihilations are

---

[6]See the discussion in Sect. 4.1.4.

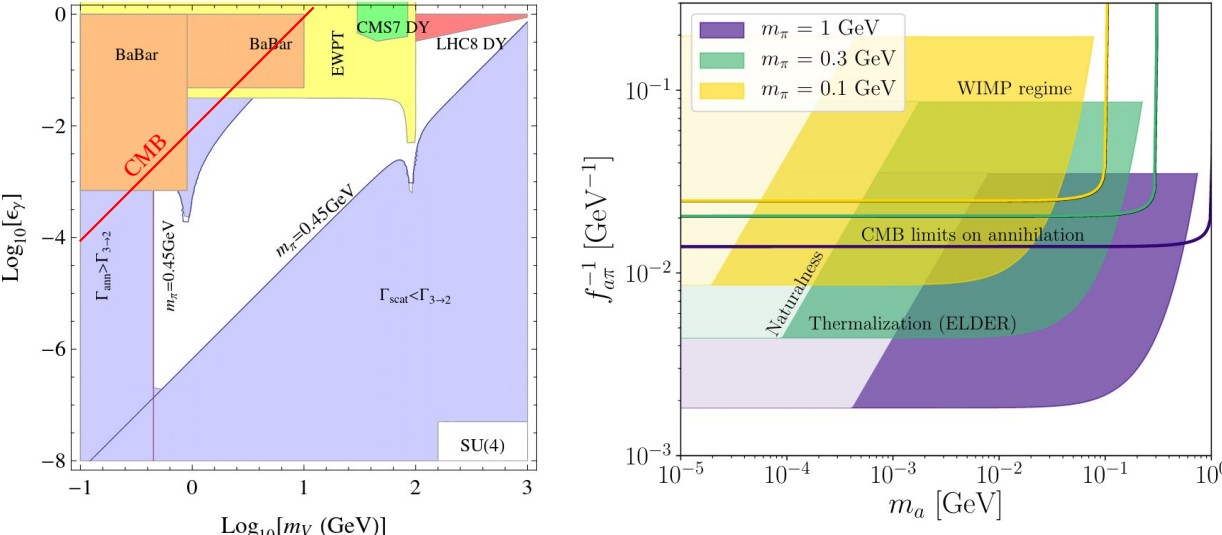

Figure 15: Left: Allowed ranges (unshaded) of kinetic mixing $\epsilon$ versus dark photon mass $m_{Z'}$ for keeping dark pions in equilibrium with the SM during SIMP production, adapted from Ref. [103]. The U(1)$'$ gauge coupling is taken to be $\alpha' = 1/4\pi$. Right: allowed regions (shaded) for axion-mediated thermalization of SIMP-produced dark pions, from Ref. [104]

suppressed by $Z'$ propagators, as well as kinetic mixing $\epsilon$, and can be subdominant to $3 \rightarrow 2$ annihilations for small enough $\epsilon$. Yet $\epsilon$ must be large enough to maintain kinetic equilibrium between the two sectors through $\pi'$-SM elastic scattering. This leads to allowed regions in the plane of $\epsilon$ and $m_{Z'}$ like in Fig. 15 (left). The CMB bounds (see for example Ref. [105]) were not considered in Ref. [103], but my estimate (red line) shows that they are less constraining than BaBar.

Another means of thermal equilibration is through an axion coupling to the dark quarks. Ref. [104] extended the earlier model of [98], noting that the $a\pi'^3$ coupling vanishes in Sp(2N) theories, avoiding semi-annihilation processes $\pi'\pi' \rightarrow \pi'a$, but the $\pi'^2 a^2$ interaction exists. Depending on the axion mass $m_a$ and its coupling to pions, $\sim (m_{\pi'}/f_{a\pi'})^2$, $\pi'\pi' \rightarrow aa$ annihilation can be subdominant to $\pi'\pi'\pi' \rightarrow \pi'\pi'$, while equilibration with the SM can be maintained if the axion-photon coupling $f_{a\gamma}^{-1}$ is large enough. Allowed regions of the $f_{a\pi'}^{-1}$-$m_a$ parameter space are shown in Fig. 15 (right). The CMB constraint is relevant here, reducing the allowed region for light $m_{\pi'} \sim 0.1$ GeV.

### 4.1.4 Role of vector mesons

Models with dark pions inevitably have heavier vector meson states $V$ as well, which can play a role in freezeout. Ref. [106] showed that vector exchange in the $3 \rightarrow 2$ annihilations can be near resonance, which allows for a higher range of possible $m_{\pi'} \lesssim 1$ GeV through the SIMP mechanism, without having to resort to nonperturbative couplings in which the chiral perturbation expansion is breaking down.

Ref. [107] observed that the semi-annihilation process $\pi'\pi' \rightarrow \pi'V$ can often dominate over $3 \rightarrow 2$ annihilation, followed by the vector decaying into SM particles. They use the same setup as in Ref. [103], but take into account the effects of the vector mesons, whose mass is expected to be $m_V \sim 4\pi f_{\pi'}/\sqrt{N}$, and can mix with the $Z'$, similar to $\rho$-$\gamma$ mixing in the SM. Generically $m_V$ could be less than $2m_{\pi'}$, in which case $V \rightarrow 2\pi'$ is blocked, and $V$ will instead decay to SM fermions through its mixing to $Z'$ and kinetic mixing $\epsilon$ of $Z'$ with the photon.

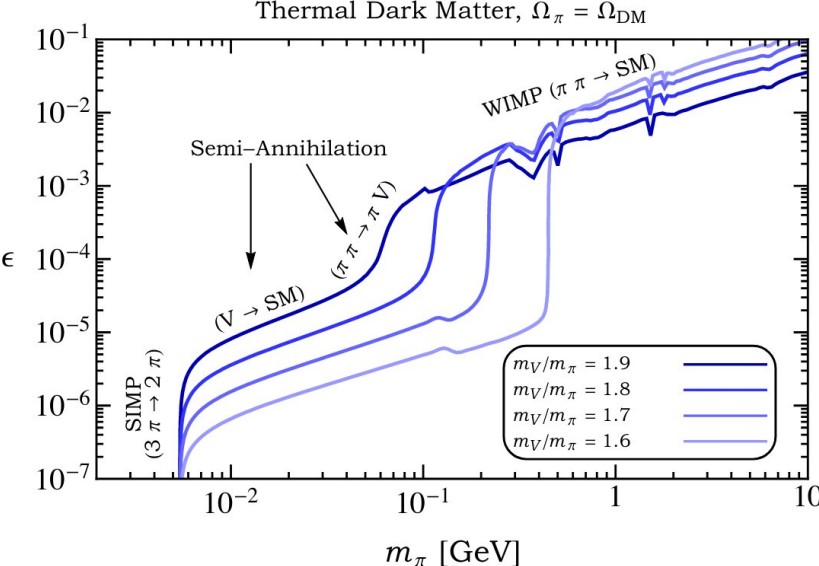

Figure 16: Required value of kinetic mixing versus $m_{\pi'}$ to obtained the observed DM density, taking account of vector meson effects, from Ref. [107].

Ref. [107] also emphasized that the cancellation of the chiral anomaly by the choice of quark charges $Q = \mathrm{diag}(1, -1, -1)$ is not sufficient for stability of $\pi'^0$, so that only the charged states $\pi'^\pm$ will be the stable DM. In the parameter space of $\epsilon$ versus $m_{\pi'}$, the $3 \to 2$ mechanism for thermal freezeout is seen to occupy a relatively small region in Fig. 16 when the effects of the vector are taken into account. At large $\epsilon$, freezeout is dominated by $\pi'^+ \pi'^- \to Z', V \to$ SM $(f\bar{f})$. The latter cross section is $p$-wave suppressed and therefore does not lead to strong CMB constraints.

## 4.2 Composite Higgs models

Composite Higgs models provide a compelling motivation for dark mesons as DM, in contrast to a secluded hidden sector. Analogously to QCD, techniquarks with an approximate flavor symmetry $\mathcal{G}$ that breaks to $\mathcal{H}$ when the confining technicolor interaction creates a techniquark condensate, give rise to pNGBs corresponding to the broken generators of $\mathcal{G}/\mathcal{H}$. Some of these should correspond to the Higgs boson, and if there are additional ones, they can be DM candidates [108].

A related example is the gauge group SU(2) with $N_f = 2$ Dirac flavors [109]. For massless quarks, this has the flavor symmetry SU(4), since each Dirac field has two chiralities, and lattice studies show it breaks to Sp(4), giving 5 Goldstone bosons. Three of these can be used for electroweak symmetry breaking (EWSB), leaving two as scalar DM candidates. This model does not actually have a composite Higgs (not obviously); this is why two rather than one of the extra Goldstone bosons are DM, and since they appear as components of a complex scalar, they can be asymmetric DM.

Another popular global symmetry breaking pattern is SO(6)→SO(5), which also has five Goldstone bosons, four of which are identified with the complex Higgs doublet, leaving one as a DM candidate.[7] Unlike the simpler minimal composite Higgs model [112] which has SO(5)→SO(4), it can be UV-completed in a techniquark setting [110]. In this model the DM

---

[7]However it is not generally stable, without additional global symmetries, due to the WZW interaction (analogous to that for $\pi^0 \to \gamma\gamma$ decay) which allows it to decay into electroweak gauge bosons. This can be overcome by taking the coset structure SO(7)/SO(6) [111]. The resulting DM is stabilized by a dark U(1) symmetry.

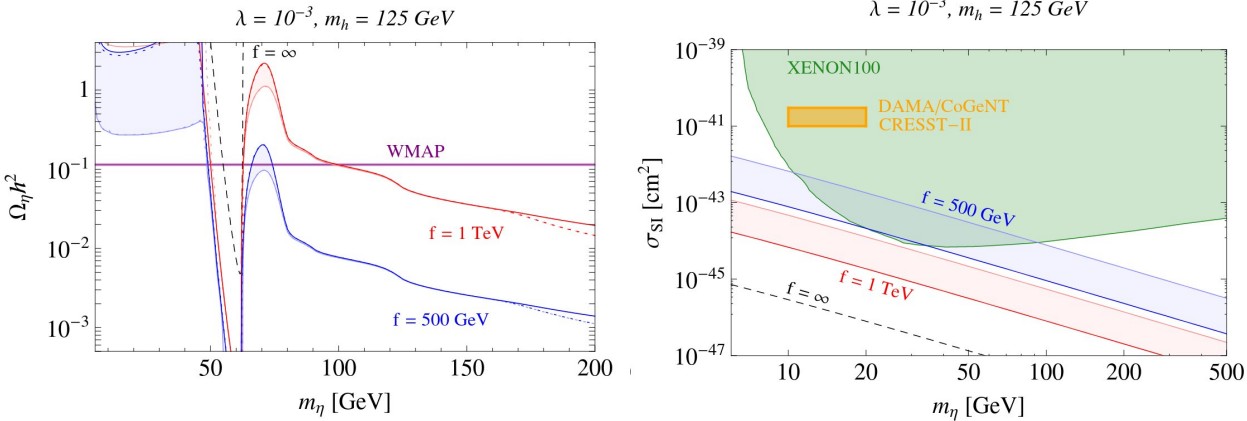

Figure 17: Predictions for the relic density and direct detection for a dark meson $\eta'$ from a SO(6)$\to$ SO(5) composite Higgs model, from Ref. [110].

meson $\eta'$ has derivative couplings to the Higgs, $\partial_\mu \eta'^2 \partial^\mu |H|^2/f^2$, standard Higgs portal couplings $\lambda \eta'^2 |H|^2$ and couplings to SM fermions, $\sim (\eta'/f)^2 y_f \overline{Q}_f H f$, that allow for $\eta'\eta' \to f\bar{f}$, $HH$ to give thermal freezeout in two different mass regimes: 50-70 GeV (with Higgs resonance from the $\lambda v h \eta'^2$ interaction dominating) and 100-500 GeV (with derivative couplings dominating), illustrated in Fig. 17. The value of the portal coupling $\lambda$ is not predicted, and direct detection rules out much wider ranges of $m_{\eta'}$ when $\lambda = 0.1$.

The same class of models was further examined in Ref. [113], focusing on LHC constraints. Searches for composite vector resonances, which mix with the SM weak gauge bosons, exclude low values of the decay constant $f < 800\,\text{GeV}$ and hence lower $\eta'$ masses. For $f = 1.1\,\text{TeV}$, $m_{\eta'} \sim 100-200\,\text{GeV}$ is predicted, as shown in Fig. 18. It is seen that indirect constraints, in this case production of antiprotons from the primary annihilation products from $\eta'\eta'$ annihilation in the galaxy, exclude much of the $\lambda$ versus $m_{\eta'}$ parameter space.

Other possible coset structures for composite dark sectors have been explored in Ref. [114], including [SU(2)$^2$×U(1)]/ [SU(2)×U(1)] and SU(3)/[SU(2)×U(1)]. Their low energy effective descriptions are inert Higgs doublet or triplet DM models, respectively. Ref. [115] studied the SU(4)×SU(4)/SU(4) model, which has 15 Goldstone bosons, and predicts composite DM with mass 500-1000 GeV.

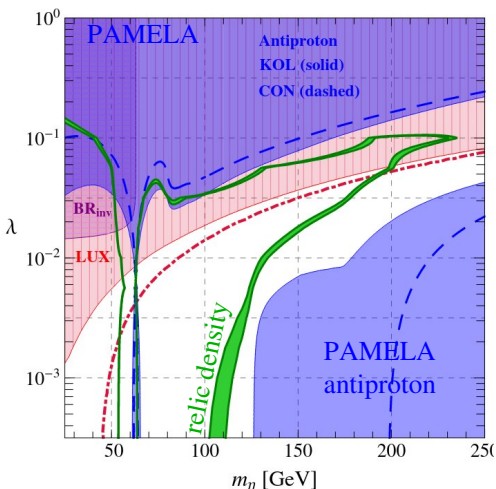

Figure 18: Combined relic density, direct and indirect detection constraints, again for the SO(6)→SO(5) model, from Ref. [113].

## 4.3 Self-interactions

One of the first motivations for dark mesons was to account for strong DM self-interactions for structure formation. Ref. [33] computed the $\pi'\pi'$ elastic scattering cross section from the chiral Lagrangian (40), with a different normalization $F_{\pi'} = 2f$ (such that $F_\pi = 93\,\text{MeV}$ for

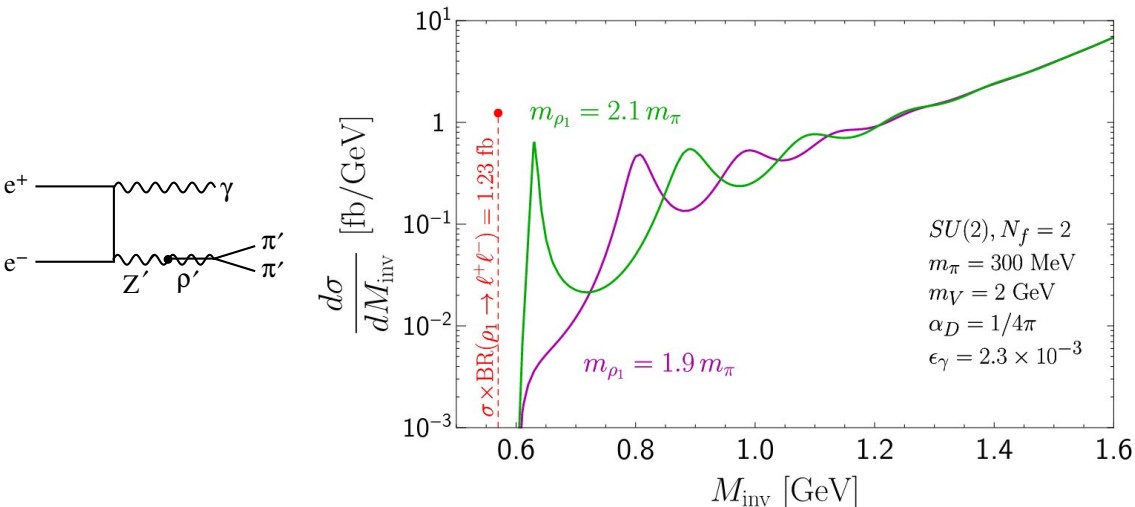

Figure 19: Left: diagram for producing a photon and invisible dark pions from $e^+e^-$ scattering. Right: predicted spectrum for invisible invariant mass, from Ref. [117].

QCD), to find

$$\sigma = \frac{m_{\pi'}^2}{32\pi F_{\pi'}^4} \left( \frac{2N_f^4 - 25N_f^2 + 90 - 65/N_f^2}{N_f^2 - 1} \right) \tag{46}$$

for $N_f$ flavors. To relate $m_{\pi'}$ and $F_{\pi'}$ to the more fundamental parameters $N$ and $\Lambda'$, lattice gauge theory calculations would be required [69, 116]. Desired values of $\sigma/m$ can be attained for a range of masses $m_{\pi'} = 30$–$100$ MeV, for $N_f = 2$–$6$. More generally, Bullet Cluster constraints put a lower bound on mesonic DM masses of this order.

Ref. [98] discusses the analogous result to (46) for the case where flavor symmetry is strongly broken by the quark masses so that there is a single lightest state that dominates the scattering, obtaining $\sigma = a^2 m_{\pi'}^2/(32\pi f_{\pi'}^4)$ (note the different normalization of $f_{\pi'}$, as in Eq. (43)), where $a \sim 2$ for SU($N$) and O($N$) gauge theories, and $a \sim 1$ for Sp($N$).

## 4.4 Detection

We have seen in the previous descriptions several examples of direct and indirect detection of dark mesons, or collider constraints on the model due to resonant production of the associated dark vectors. Because of its scalar nature, the Higgs portal is a common interaction for dark mesons, which leads to scattering on nuclei by Higgs exchange [119, 120]. Composite Higgs models can also have direct dimension-6 couplings of the dark meson to SM fermions [110, 115]. Light metastable dark mesons that can decay to electrons or photons are constrained by the CMB to have lifetimes $\tau \gtrsim 10^{25}$ s [121].

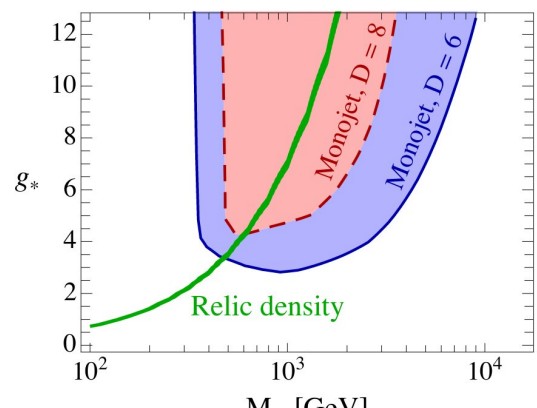

Figure 20: from Ref. [118]; see text.

A distinctive signal of dark mesons, "SIMP spectroscopy," was suggested in Ref. [117] for $e^+e^-$ collisions. The Feynman diagram

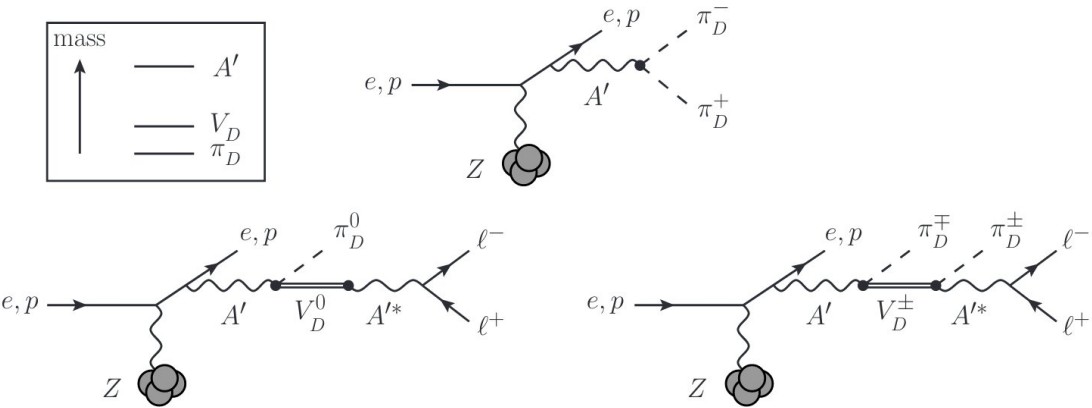

Figure 21: Novel signals from dark mesons coupled to dark photons for fixed target experiments, from Ref. [107].

is shown in Fig. 19 (left): it produces a visible photon and invisible dark pions, through the kinetically mixed $Z'$ portal. The $Z'$ mixes with the dark vector meson $\rho'$ to produce $\pi'\pi'$. Through the kinematics, the invariant mass of the invisible particles $M_{\text{inv}}$ is determined by the beam energy $\sqrt{s}$ and the observed photon energy: $M_{\text{inv}}^2 = s - 2E_\gamma\sqrt{s}$. The spectrum of vector excitations, expected to go as $m_{\rho_n}^2 \sim 4n$ in an AdS-QCD approach [122], can be observed through the resonances in the differential cross section $d\sigma/dM_{\text{inv}}$, an shown in Fig. 19 (right).

Ref. [107] emphasized the opportunities for fixed-target experiments to observe similar novel effects connected with production of dark mesons with interactions to a light (below 10 GeV) $Z'$ with kinetic mixing $\epsilon$ and mixing with the dark vector excitations. These processes are illustrated in Fig. 21. Searches for these signals will be able to probe currently allowed regions of the $\epsilon$-$m_{Z'}$ plane by planned future experiments.

An interesting example of complementarity between the relic density requirements and detection at colliders was discussed in Ref. [118]. Under the assumption that the new confining dynamics respects approximate SM symmetries, including custodial, flavor, baryon and lepton number, and can be described by a single new scale $M$ and coupling $g_*$ [123], the leading dimension-6 and 8 couplings of mesonic DM $\pi'$ to the SM can be parametrized up to order 1 coefficients; for example, operators like

$$\frac{g_*^2}{M^2}|\partial_\mu\pi'|^2|H|^2, \quad \frac{g_*^2}{M^2}\partial_\mu\pi'^*\partial_\nu\pi' B^{\mu\nu}. \tag{47}$$

The requirement of a thermal relic density fixes $g_*$ in terms of $M$ as shown in Fig. 20, where $m_{\pi'} = 5\,\text{GeV}$ was assumed. The shaded regions are excluded by ATLAS searches for monojets [124], putting an upper bound on the scale of confining dynamics $M \lesssim 500\,\text{GeV}$ in this example.

Inelastic DM scattering in direct searches requires very small mass splittings $\lesssim 100\,\text{keV}$, that can be naturally achieved in composite models. Ref. [125] used the small hyperfine splitting between a dark scalar and vector meson $\pi_d$ and $\rho_d$ in an SU(N)$'\times$U(1)$'$ sector with kinetic mixing to construct such a scenario.

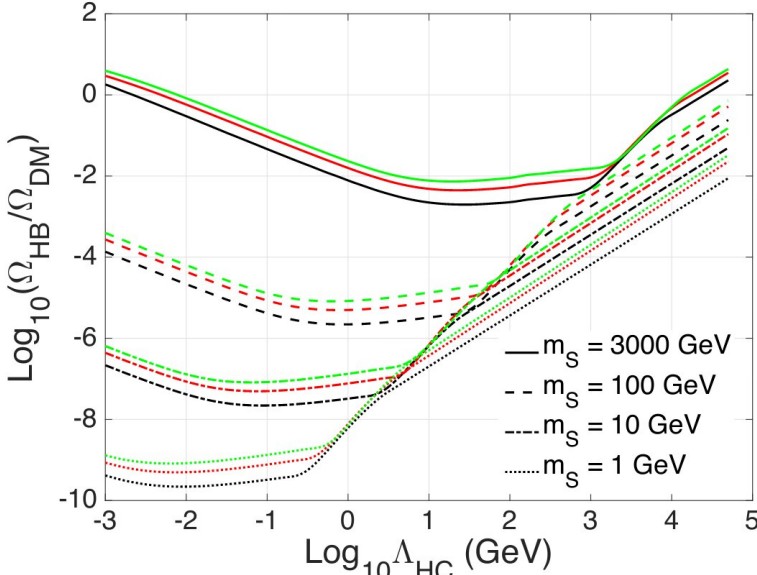

Figure 22: Relic density of dark "hyperbaryons" versus the "hypercolor" confining scale, for a series of constituent quark masses $m_S$, and number of colors $N = 2, 3, 4$ (blue, red and green curves, respectively), from Ref. [127].

# 5 Dark baryons

The mass density of the visible universe is dominated by baryons, so the possibility of dark baryons as DM seems particularly natural. For SU(N) or SO(N) theories, these would be $Q^N$ bound states of the dark quarks $Q$, whose spin could be $N/2$ (if all $Q$'s are of the same flavor) or possibly lower (if there are several flavors). In the SU(N) case they are complex, admitting the concept of conserved dark baryon number, while for SO(N) they are real, but can nevertheless still be stable [126]. Ref. [69] notes an advantage of dark baryons: even if they are not stable, their decays will be mediated by operators of dimension $d \geq 6$ if $N \geq 3$. This makes them more easily long-lived on cosmological timescales.

In the SM, quark masses are much less than $\Lambda_{QCD}$, making it difficult to compute the detailed properties of baryons. One can use results from lattice gauge theory to infer some of the properties of dark baryons in the case of SU(3) [116]. A computationally simpler regime is where $M_Q \gg \Lambda'$. Then the quarks are nonrelativistic, and their masses and binding energies can be calculated using familiar quantum mechanical techniques for nonrelativistic bound states. The details of freezeout are different in these two cases, as we will discuss.

## 5.1 Relic density

Like for dark atoms, there is a model-dependent issue as to whether the dark baryons have an asymmetry or not. Independently of this issue, one can address whether their symmetric component can be large enough to account for all of the DM in a more model-independent way. In the case $m_Q \ll \Lambda'$ analogous to QCD, one could expect that the cross section for $p\bar{p}$ annihilation scales with the baryon mass as in QCD, $\sigma v \sim 100/m_B^2$. Matching this to the usual cross section for thermal freezeout [128], one finds that $m_B \sim 200\,\text{TeV}$ [129], close to the unitarity limit [93, 94]. In this case $\Lambda'$ is above the freezeout temperature of the baryons, $\sim m_B/25$, so the details of the confining transition are not important.

In the opposite case of heavy quarks, $m_Q \gg \Lambda'$, annihilation of $Q\bar{Q}$ occurs before hadronization, and the details of hadronization are affected by the residual $Q$ density. The qualitative

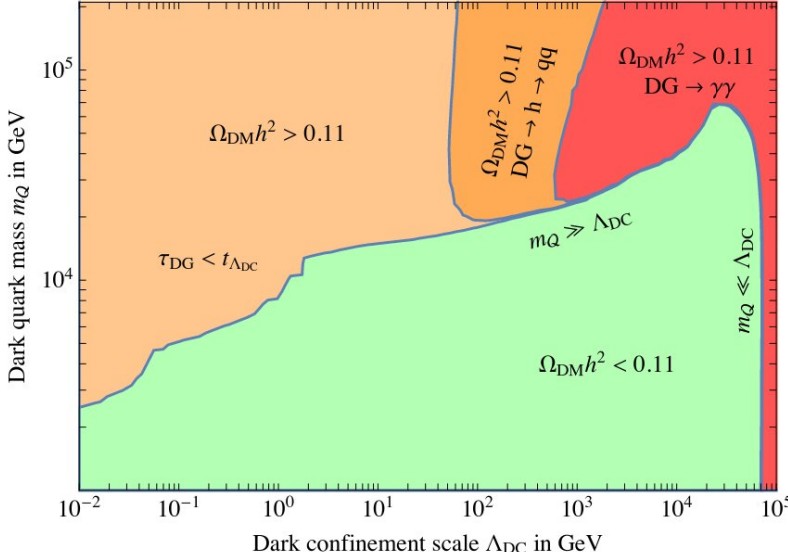

Figure 23: Regions of dark quark mass $m_Q$ versus confinement scale $\Lambda'$ relevant for the relic density of dark baryons, for gauge group SU(3), from Ref. [130]. Observed abundance is along the boundary of the green region.

difference between the two scenarios can be seen in Fig. 22 [127], which solved the Boltzmann equation in the general case. The power-law scaling for $\Lambda' > m_Q$ reflects the standard relation $\Omega \sim 1/\langle \sigma v \rangle \sim \Lambda'^2$ for thermal freezeout: baryons form at an early time, and their final abundance is independent of initial conditions at the confinement temperature, and only mildly dependent on $m_Q$. For $\Lambda' < m_Q$, this scaling breaks down because the initial density of baryons, formed at the confinement transition, is much higher than their equilibrium abundance at that temperature, since $m_B \sim N m_Q$, and so

$$n_{Q,\text{eq}} \sim e^{-m_Q/T} \gg n_{B,\text{eq}} \sim e^{-N m_Q/T} \, . \tag{48}$$

Initially $n_B \sim n_Q/N \gg n_{B,\text{eq}}$ from hadronization at $T' \sim \Lambda'$. As a result the relic density has a more complex dependence on $m_Q$ and $\Lambda'$, sensitive to the dark baryon density at $T' \sim \Lambda'$. We assumed a geometric cross section for $B$-$\bar{B}$ annihilation into dark pions, with a size determined by solving the nonrelativistic bound state problem with an appropriate potential, and assumed portals for keeping the two sectors in equilibrium.

These results show for $m_Q \gg \Lambda'$, $m_Q$ above the TeV scale is favored for getting the observed DM abundance. This was further explored in Ref. [130], which took into account the effects of the dark glueballs that inevitably also form. If they are long-lived enough to temporarily matter-dominate the universe, their decays to DM particles will dilute the $B$ abundance. The favored values of $m_Q$ versus $\Lambda'$ are shown in Fig. 23 (boundary of green region).

The two regimes can also be described in terms of weakly coupled baryons, $m_Q \gg \Lambda'$, in which glueballs provide the thermal bath of the dark sector, or strongly coupled, $m_Q \ll \Lambda'$, where pions play that role [132]. The allowed regions for the relic density can be expressed in the parameter space of $m_B$ versus $m_{\pi'}$ in the former case, and in terms of $m_B$ versus the glueball mass in the latter. The results are sensitive to whether the particles in the bath are sufficiently long-lived to cause entropy dilution of the baryons by their decays, and whether they heat up due to $3 \to 2$ interactions. The latter effect can make the dark sector temperature higher than that of the SM, and the relic $B$ density is enhanced by a factor of $T'/T$ or $(T'/T)^{3/2}$, depending on which sector is dominant. This requires larger-than-normal annihilation cross sections for

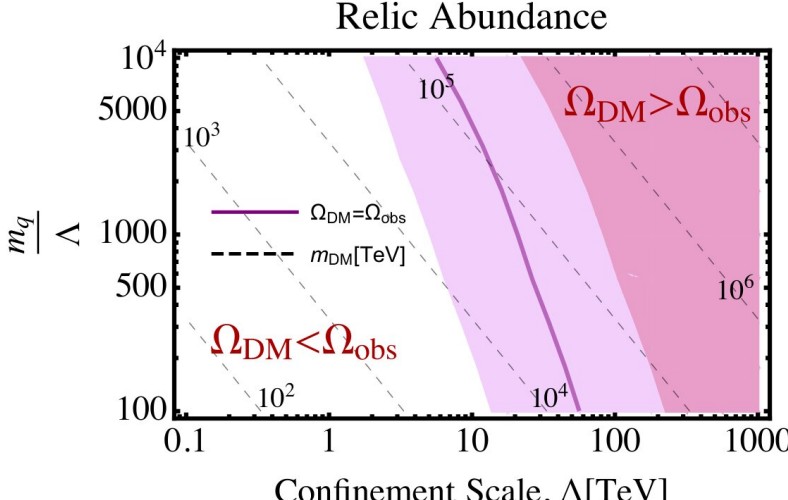

Figure 24: Regions of $m_Q/\Lambda'$ versus confinement scale $\Lambda'$ for the relic baryon abundance, accounting for bubble dynamics from the first order confinement transition, from Ref. [131]. Dashed lines show contours of constant $m_B$.

getting the right $B$ abundance, which in turn enhances the signals for indirect detection from annihilation in the galaxy, despite the large $m_B \gtrsim 10\,\text{TeV}$.

However, this is not the end of the story, for the regime where $\Lambda' \ll m_Q$, since the SU(N)$'$ gauge theory is known to have a first order confinement transition for $N \geq 3$, and the nucleation of bubbles can play an important role. Refs. [131, 133] show that the quarks are kept outside of the bubbles of confined phase because of the energetic cost of having a free quark. After the bubbles percolate, the quarks get squeezed into small pockets of residual deconfined phase, where they mostly annihilate away. But since in each such pocket there is a statistical $\sqrt{N_q}$ imbalance ($N_q$ being the number of quarks in the pocket) between quarks and antiquarks, some small asymmetry is guaranteed to remain, that hadronizes into baryons, which can then escape to the confined phase. This leads to a much smaller yield of dark baryons than in the previous estimates, as shown in Fig. 24. Instead of $m_B \sim 100\,\text{TeV}$, values of $10^4$–$10^5\,\text{TeV}$ are needed to get the observed abundance. This suppression is only ameliorated in the $\Lambda' \gtrsim m_Q$ regime, where the phase transition weakens into a smooth crossover [134]. Although this process is dubbed "accidentally asymmetric dark matter," there is no global asymmetry, since the sign of the asymmetry from each pocket is random. As for any such model with conserved dark baryon number, the strong constraints can be circumvented by introducing a primordial asymmetry. Another loophole is the SU(2) case [135], which has a second order transition [136].

## 5.2 Self-interactions

The elastic scattering cross section for baryons can be estimated by large-$N$ and NDA to be of order

$$\frac{\sigma_{BB}}{m_B} \sim \frac{4\pi}{N\Lambda'^3} \tag{49}$$

since $\sigma_{BB} \sim 4\pi/\Lambda'^2$ [82, 137] and $m_B \sim N\Lambda'$. Witten showed that the amplitude for $B$-$B$ scattering scales as $\mathcal{M} \sim N$, but the cross section is $\sigma_{BB} \sim |\mathcal{M}|^2/m_B^2$, so the factors of $N$ cancel out. Comparing to the actual value of $\sigma_{pn}$ in QCD, the estimate (49) is too small by a factor of 50. (I choose proton-neutron scattering here so that the Coulomb interaction that would become relevant at low energies for $pp$ scattering is not an issue.) It turns out that the cross section is resonantly enhanced by the weakly bound deuteron ($np$) state, whose binding

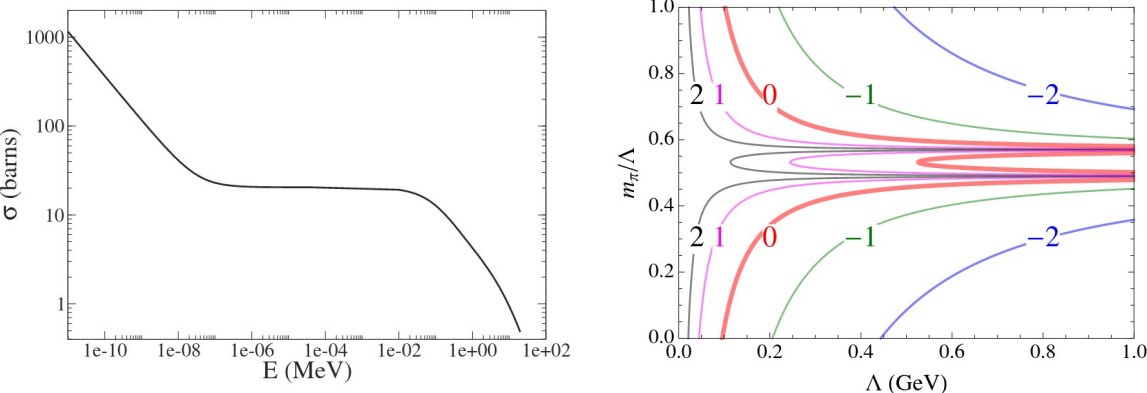

Figure 25: Left: actual neutron-proton scattering cross section as a function of center of mass energy. Plateau is the constant cross section of Eq. (50); at lower energies the electromagnetic interaction dominates. Right: contours of $\log_{10}[\sigma/m]/[0.6\,\text{cm}^2/\text{g}]$ in the plane of $m_\pi/\Lambda$ versus $\Lambda$ for SU(3) gauge theory, from Ref. [33].

energy is $E_b = 2.2\,\text{MeV}$, and $\sigma \sim 2\pi/(\Lambda E_b)$ gives a better estimate of the cross section. The point is that another scale $E_b$ is appearing in the problem, that cannot be anticipated from order-of-magnitude arguments.

Ref. [33] noted that one can make quantitative predictions, for the case of SU(3), by appropriating results from lattice gauge theory [138]. Lattice gauge theory is computationally expensive for light quarks, so it is typical for simulations to be done with a series of decreasing quark masses, for extrapolation to realistically small values. This study of the dependence of observables on varying quark masses can be valuable to the composite model builder. For the present case, nucleon scattering amplitudes in the spin singlet and triplet channels (correlated by Fermi statistics with the isospin channels) were determined as a function of the pion mass (related to $m_q$ by $m_\pi \sim (\Lambda m_q)^{1/2}$), and expressed as scattering lengths $a_s$ and $a_t$, defined by the relation

$$\sigma = \pi(a_s^2 + a_t^2) \tag{50}$$

as $v_{\text{rel}} \to 0$. We fit to the results of [138] to approximate

$$a_s\Lambda' \cong \frac{0.58}{m_{\pi'}/\Lambda' - 0.57}, \quad a_s\Lambda' \cong \frac{0.39}{m_{\pi'}/\Lambda' - 0.49}, \tag{51}$$

where the poles indicate the values of $m_{\pi'}/\Lambda'$ at which a bound state in the $np$ or $nn/pp$ channel is just starting to appear. Combining Eqs. (50,51) allows one to engineer dark sectors where the low-velocity baryon self-interactions would match a desired cross section for small-scale structure problems. This is illustrated in Fig. 25 (right), where the thick curves correspond to a constant cross section of $0.6\,\text{cm}^2/\text{g}$. This does not take advantage of the velocity-dependence at high energies to fit galactic cluster profiles versus smaller halos [139, 140], which might be worth investigating.

## 5.3 Direct detection

A fully secluded hidden sector is safe from direct detection, but one often prefers to assume there is a portal to the SM to maintain thermal equilibrium, since this facilitates thermal freeze-out, and of course it is more interesting to detect DM than not to detect it. The dark baryon is typically a bound state of quarks $Q$ that are singlets under the SM gauge symmetries, although electroweak triplets or quintuplets (with vanishing hypercharge) are viable possibilities [126].

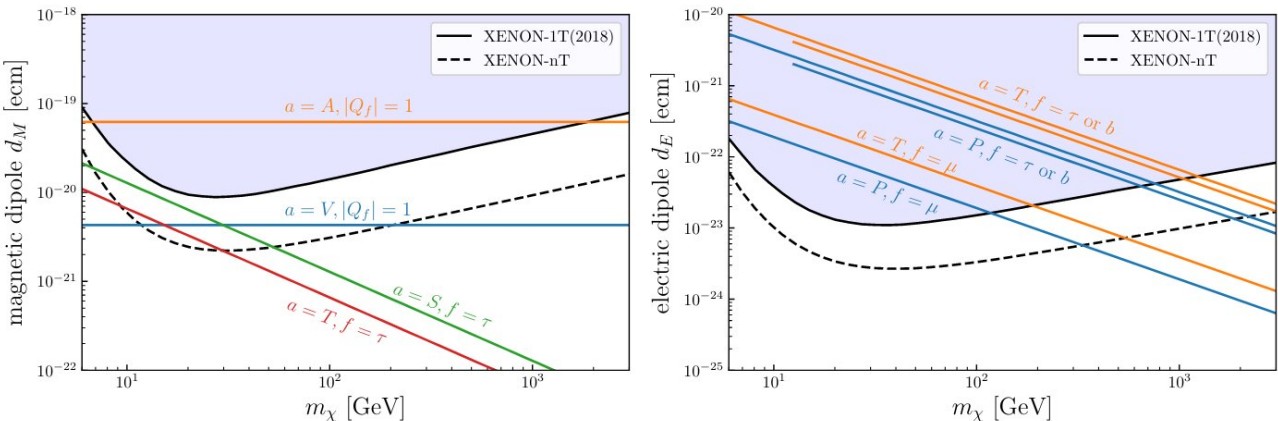

Figure 26: Direct detection constraints on dark magnetic dipole moments (left) and electric dipole moments (right), from Ref. [141].

It is also possible to have doublet $Q$ if there is a custodial symmetry that prevents weak neutral current interactions with the baryons [142], dubbed "stealth" DM.

Even if $Q$ is a singlet, if it has interactions with charged particles, *e.g.,*

$$\lambda \overline{Q} \Phi \psi \,, \tag{52}$$

where $\overline{Q}\Phi$ is neutral under $SU(N)'$ and $\Phi\psi$ is electrically neutral, but $\Phi$ and $\psi$ are electrically charged, then $Q$ and hence its associated baryon $B = Q^N$ acquire a magnetic moment at one loop, which is subject to direct constraints. In more complicated models, an electric dipole moment could be generated.

Similarly if $Q$ is an electroweak triplet and $\psi$ is a doublet, both fundamental under $SU(N)'$, the interaction

$$\lambda \overline{Q}^i H^\dagger \tau_i \psi \,, \tag{53}$$

(showing the $SU(2)_{EW}$ index) leads to mass mixing between $Q$ and $\psi$ when the Higgs gets a VEV. The mass eigenstate $Q'$ thus couples to $H$. Even without the interaction (53), the neutral $T_3 = 0$ component of the triplet and quintuplet models, which is the DM candidate, gets a one-loop coupling to the nucleons, with the charged components and $W^\pm$ in the loop [143]. (The DM coupling to $Z$ vanishes if $Q$ has no hypercharge.) Another generic possibility is for $Q$ to be charged under $U(1)'$ that is kinetically mixed with hypercharge.

In the stealth model [142], two flavors of vector-like hyperquarks with even $N \geq 4$ number of colors are introduced, where $Q = (u, d)_L$, $u_R$, $d_R$ are doublet and singlets respectively under $SU(2)_{EW}$. This allows for Higgs couplings to the hyperquarks. Custodial $SU(2)$ symmetry forbids neutral weak current interactions of the baryons $B$, and their even number of constituents forbids magnetic moments, leaving the Higgs portal as the only means of detection.

### 5.3.1 Magnetic and electric dipole moments

If $N$ is odd, then the $B = Q^N$ baryon can have nonvanishing spin, which is a necessary requirement for having an electric or magnetic dipole moment. Using the quark model, we would estimate that the baryon dipole moment is $\mu_B \cong N\mu_Q$ (or less if there are several flavors of quarks and their spins do not all add). The one-loop contribution to the magnetic moment (MDM) $\mu_Q$ from the interaction (52) is of order [144]

$$\mu_Q \sim \frac{e\lambda^2 m_Q}{64\pi^2 M^2} \,, \tag{54}$$

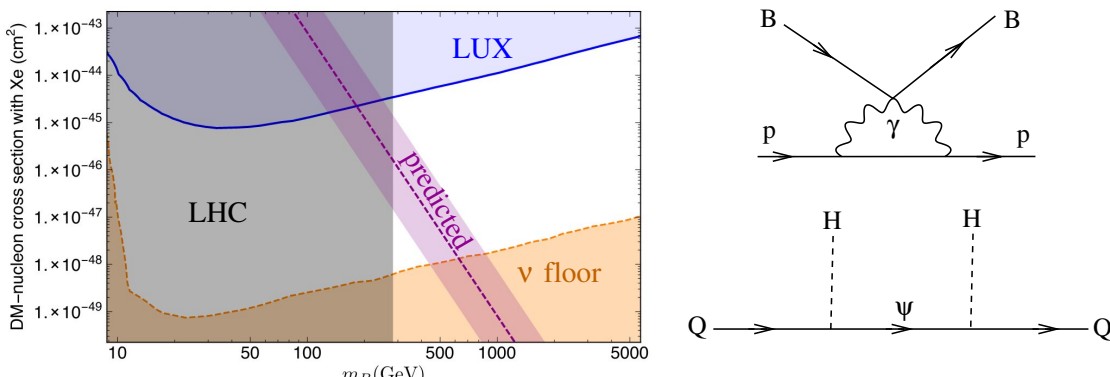

Figure 27: Left: constraints on stealth DM (spin 0 baryon) scattering on protons induced by electric polarizability, adapted from Ref. [147]. Upper right: dark baryon-proton scattering induced by electric polarizability operator of Eq. (55). Lower right: diagram leading to effective Higgs portal coupling to a hyperquark $Q$ that is a triplet under $SU(2)_{EW}$.

where $M$ is the largest mass in the loop. In a more complicated theory having several complex couplings with unremovable phases, the loop diagrams could also give rise to an electric dipole moment $d_Q$, in analogy to the contribution of a CP-violating $\pi NN$ coupling to the neutron EDM [145]. This occurs in composite Higgs models (technicolor) [126]. The EDMs are more strongly constrained than the MDMs because their cross section for scattering on nucleons is enhanced by a factor of $1/v_{\mathrm{rel}}^2$ compared to that of MDMs [146]. Recent constraints on dark MDMs and EDMs from direct searches are shown in Fig. 26. For example taking $SU(3)'$ and $m_B = 2\,\mathrm{TeV}$, Eq. (54) implies $\lambda/M \lesssim 2/\mathrm{TeV}$.

In models with even $N$, even though there are no dipole moments, there can exist higher dimension couplings to photons—polarizability, which for a scalar baryon $B$ takes the form

$$C_F |B|^2 j_\mu F^{\mu\alpha} F_{\alpha\nu} j^\nu \tag{55}$$

in an external current $j_\mu$. Interactions of $B$ with protons then occur at one loop (Fig. 27 (upper right)). Lattice predictions for the polarizability $C_F$ for SU(4) and ensuing constraints from direct searches were carried out in Ref. [147]; see Fig. 27.

### 5.3.2 Higgs portal

For the $SU(2)_{EW}$ triplet $Q^i$ model, the interaction (53) in the diagram Fig. 27 (lower right) leads to the effective Lagrangian

$$\begin{aligned}
\mathcal{L} &= \frac{\lambda^2}{m_\psi} \overline{Q}^i H^\dagger (\delta_{ij} + i\epsilon_{ijk}\tau_k) H Q_j \\
&\rightarrow \frac{\lambda^2 v}{m_\psi} h \overline{Q} Q \equiv y_{\mathrm{eff}} h \overline{Q} Q,
\end{aligned} \tag{56}$$

where $v = 246\,\mathrm{GeV}$ is the Higgs VEV and we assumed $m_\psi \gg m_Q$ (appropriately, since we want $Q$ to be the dark matter). To determine the $B$-nucleon scattering cross section, we need the matrix element

$$\langle B | \overline{Q}^i Q_i | B \rangle \equiv f_B, \tag{57}$$

where the form factor $f_B$ is a number or order 1 (or perhaps $N$). The cross section for scattering on nucleons from Higgs exchange is

$$\sigma_{BN} = \frac{(y_{\text{eff}} f_B f_N)^2 m_N^4}{2\pi v^2 m_h^4}, \tag{58}$$

where $f_N \cong 0.3$ for the Higgs-nucleon form factor. The XENON1T [148] (PandaX-4T [149]) limits are

$$\sigma \lesssim 10^{-48(-48.4)} \left(\frac{m_B}{\text{GeV}}\right) \text{cm}^2 \tag{59}$$

for DM mass $m_B$ in the high-mass region, giving $y_{\text{eff}} < 0.06(m_B/\text{TeV})^{1/2}$. This gives $\lambda \lesssim 0.1$ for $m_B \sim 1$ TeV, $m_\psi \sim 500$ GeV, for example.

### 5.3.3 Kinetic mixing portal

If the quarks couple to a massive, kinetically mixed $Z'$ with charge $g'$, the baryon has charge $Ng'$, and conservation of the vector current implies that $Z'$ couples to the current $Ng'\bar{B}\gamma^\mu B$. We can use the coupling (22) to protons to compute the $p$-$B$ cross section, assuming that $m_A'$ is much greater than the momentum transfer, as

$$\sigma_{pB} = \frac{(\mu_{pB} N g' \epsilon e)^2}{\pi m_{A'}^4}, \tag{60}$$

where $\mu_{pB} = m_p m_B/(m_p + m_B) \cong m_p$ is the reduced mass. Using the experimental limit (59) gives

$$Ng'\epsilon \lesssim 10^{-8} \left(\frac{m_A'}{\text{GeV}}\right)^2 \left(\frac{m_B}{\text{TeV}}\right)^{1/2}. \tag{61}$$

### 5.4 Masses and wave functions

Frequently one would like to relate the dark baryon mass and size to the fundamental parameters of the model. For relativistic bound states this would be most reliably done using lattice gauge theory. For nonrelativistic or mildly relativistic systems, one can use quantum mechanics with a model for the two-body potential between constituents [125, 127, 130].

If $m_Q \gg \Lambda'$, the consituents are highly nonrelativistic, and the quark-quark force is dominated by the short-distance Coulomb contribution [150]

$$V_C = -\frac{\alpha'}{2r}\left(N - \frac{1}{N}\right). \tag{62}$$

For faster and hence less deeply bound quarks, the linear confining part of the potential can become significant,

$$V_L = \sigma r \cong 2(N-1)\Lambda'^2 r, \tag{63}$$

where the string tension $\sigma$ was estimated by Ref. [127] using large-$N$ scaling together with lattice determinations for $N = 3$.

To approximately solve the Schrödinger equation for the full potential,

$$V = Nm_Q + \sum_{i<j}^{N}\left[V_C(r_{ij}) + V_L(r_{ij})\right], \tag{64}$$

one can make an ansatz for the ground-state wave function

$$\psi \propto \exp\left(-\mu \sum_i^N r_i\right) \tag{65}$$

(considering $r = 0$ to be the centroid of the baryon) and use the variational method: compute the total energy $E = \sum_i p_i^2/2m_Q + V$ as a function of $\mu$ and minimize it to find the mass and size $\mu^{-1}$ of the bound state. This method can be extended to relativistic systems by using the kinetic energy $T = \sum_i \sqrt{p_i^2 + m_Q^2}$ [151] and working in the momentum basis to evaluate its expectation value.

# 6 Conclusion

Although Occam's razor seemingly makes composite dark matter not the theorist's first choice, nature may well think differently. The fact that we are made from composite (visible) matter gives credence to the possibility of a rich dark sector including gauge interactions. Whether abelian or nonabelian, this can give rise to bound states forming the dark matter.

In terms of experimental motivation, the hints of strong DM self-interactions for solving the small-scale structure formation problems of standard CDM are perhaps the most persuasive indication that DM could be composite. It is intriguing that the target cross section of $\sigma/m \sim 1\text{b/GeV}$ is of a similar order of magnitude to that for nucleons. One day experiments may provide definitive evidence that will allow us to narrow the currently vast scope of our speculations.

# Acknowledgements

I thank S. Caron-Huot, M. Fairbairn, E. Hardy, H.-M. Lee, G. Moore, D. Morrissey, E. Neil, J.-S. Roux, F. Sannino, K. Schutz, T. Slatyer, N. Toro and A. Urbano for helpful correspondence, and S. Heeba for correcting an error. I thank M. Cirelli and I. Masse for their encouragement to attend Les Houches in person in 2021. Thanks also to N. Selimovic for proofreading and helpful suggestions to improve the clarity. I thank the referees for numerous constructive suggestions.

**Funding information** This work was supported by the Natural Sciences and Engineering Research Council (NSERC) of Canada.

## A   Road bike rides from Les Houches

Mountain bikers have several options near Les Houches, but road bikers have just two: up the valley or down the valley. In either case, be prepared for significant climbing. In the direction of Vallorcine, Col des Montets is the high point and makes for a pleasant climb, especially coming back from the other side, which is very scenic. Chamonix now has a nice bike path crossing most of the town, that allows one to get off the main road (D243) and away from the traffic. After crossing the Arve on D243, take the first right to cross it again and join the bike path. After some kilometers it ends abruptly; turn left to join D1506 toward Argentière. From the Col, one can continue to Switzerland if desired. Vallorcine is a charming destination, just a few kilometers from the border.

To go down-valley, ignore Google's suggestion to take All. des Diligences (suitable for hikers and mountain bikes only); instead take Route de Vaudagne to Route de la Plaine Saint-Jean (D13). Continue to Passy and turn right at D43 (or turn a bit sooner at Chem. de Perrey toward Maffray and take Route de Maffrey to join D43) for the climb to Plaine Joux, a popular cycling challenge, where the maximum grade is marked at each kilometer, going up to 8%. Save some energy for the ride back, which entails about the same gain in elevation and similar grades.

Not highly recommended: D143 toward Lac Vert (ends at parking, far from the lake, steep grades); and especially not Parc de Merlet: very crowded, very steep, no rewarding views.

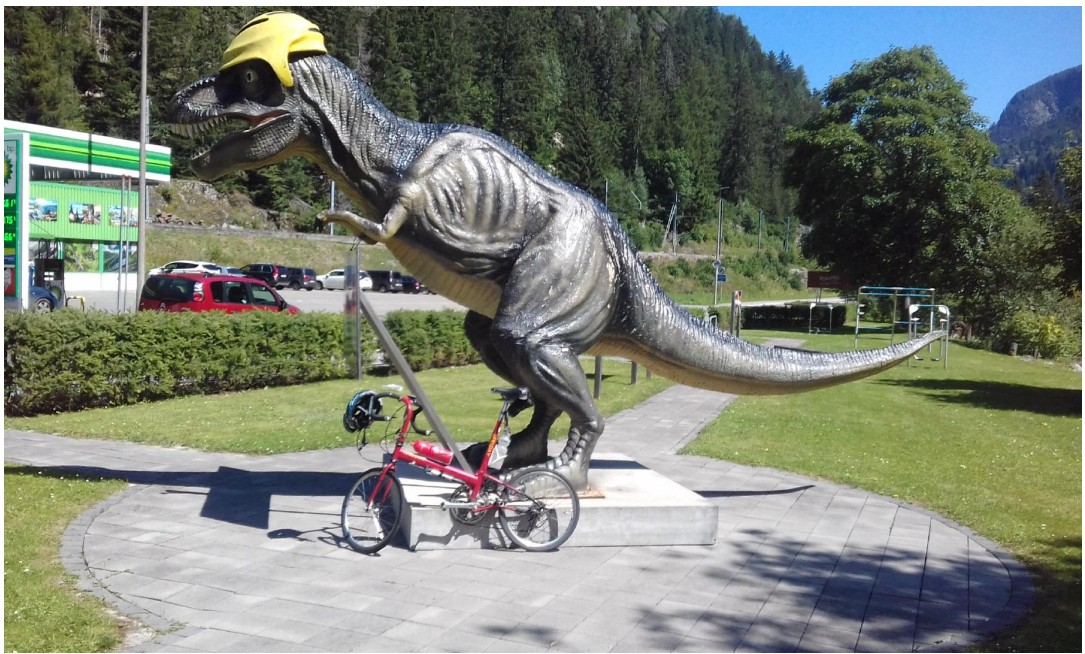

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
