# Peer review of "Dark atoms and composite dark matter"

_SciPost Physics Lecture Notes, doi:SciPost Phys. Lect. Notes 52 (2022)_

## Round 2 · Referee Report · Anonymous · 2021-10-9

Strengths
1-Clearly written
2-Comprehensive and well-referenced
Weaknesses
1-Some of the figures are not fully explained (but this doesn't matter too much since the reader can easily follow the original reference which is always provided)
Report
These lecture notes survey the landscape of composite dark matter models.
They are very well written and the discussion is clear and pedagogical.
I recommend this work for publication in SciPost Physics Lecture Notes.
I only have a few very minor questions and suggestions which I list below:
* In the second paragraph below Sec. 2.1 -- recombination occurs at a temperature
parametrically lower than the B_H' because of the small baryon density
compared to photons.
* Does the dark atom model in Secs. 2.1-2.3 rely on the absence of a symmetric DM component?
If so, are the lower bounds on the U(1) gauge coupling from the requirement
of complete annihilation interesting for any of the masses considered?
For example if I compare a naive estimate of ppbar -> gamma gamma
annihilation cross-section with Eq. 13 in https://arxiv.org/pdf/1111.0293.pdf
I find a lower bound on alpha that is potentially interesting.
This might be worth a small note, but I leave it up to the author.
* Below Eq. 23, partial derivatives are missing in the phi terms of the A/A'
gauge transformation expressions
* When discussing small scale problems of CDM which have been traditionally
used a strong motivation for DM models with significant self-interactions
it is important to acknowledge some recent work that indicates that some of
these issues are being resolved in Lamda CDM, see, e.g. https://arxiv.org/abs/1711.06267
* When discussing the lower mass bounds on WDM in the context of, e.g., dark
glueballs, it may be worth mentioning that competitive constraints are
obtained not just from Lyman alpha but also from, e.g., MW satellite counts
(https://arxiv.org/abs/2101.07810) and sub-halo mass function (https://arxiv.org/abs/1911.02663).
* Typo in the caption of Fig. 12 -- the Wilson coefficient should scale like 1/M^4
* Above Eq. 45 -- should m_S be m_Q?

---

## Round 2 · Referee Report · Anonymous · 2021-10-24

Report
The paper provides a comprehensive review of models of atomic dark matter and of models in which the dark matter candidate is a bound state composed of strongly coupled constituents. I expect that it will be of great utility to researchers looking for an introduction to these topics and a guide to the existing literature. I am happy to recommend the paper for publication. I have a few suggestions that the author may want to consider.
During the process of galaxy formation, atomic dark matter is expected to undergo shock heating as it falls into the gravitational well of the galaxy. If the resulting temperature is high enough, this could potentially reionize the dark matter gas, with the result that in the present day it is still primarily in the form of ions rather than atoms. This has been considered, for example, in Ref. [43] and in arXiv:2104.02074. The author may want to discuss this effect.
The author says that the ionized components of dark atoms, if millicharged, are expelled from the galactic disc by supernovae shock waves, citing Ref. [38]. This would have important implications for direct detection. However, I believe the result of [38] may not be applicable because the atomic dark matter model contains a dark photon, which provides an alternative path for cooling. This has been considered, for example, in arXiv:1011.5078. The author may want to clarify this point.
In the context of composite Higgs models, the author discusses the SO(6)/SO(5) symmetry breaking pattern, which leads to 5 Goldstone bosons. Four of the Goldstones are identified with the Higgs doublet of the Standard Model, while the last Goldstone is a dark matter candidate. However, it is important to note that this last Goldstone is not in general stable, unless additional symmetries are present. In particular, this symmetry breaking pattern in general admits a Wess-Zumino-Witten term that allows this particle to decay to electroweak gauge bosons. This problem can be avoided by going to SO(7)/SO(6), as discussed in arXiv:1707.07685. The author may want to mention this.

---

## Round 2 · Referee Report · Anonymous · 2021-11-9

Strengths
well organized and clear
Weaknesses
not too many technical details
Report
The manuscript is a summary of lectures given at the Les Houches Summer School 2021 about dark atoms and composite dark matter. This is a very pedagogical review of the topic and covers most of the basic ingredients. The review covers simple dark atoms, dark glueballs, dark meson and baryons. For each class of models it reviews the dark sector structure (like masses, self interactions) and the mechanism that set the relic abundance and possible signatures like direct detection, CMB and slow decay to standard model~(SM) particles.
The lecture notes are accessible to PhD students with a background on QFT, SM and dark matter. Although, the notes give minimal amount of technical calculation, the important references are pointed such that the reader can fill the missing details or further read by his/her own.
Overall, these lecture notes give a very good starting point to the topic for composite dark sector and dark atoms.

---

## Round 4 · Author Response

Thanks to the referees for helpful comments and suggestions. I have implemented them as described below.

---

## Round 4 · List of Changes

Above eq. (3), clarified that dark recombination occurs below the binding energy, due to small abundance of baryons.
Added paragraph at end of section 2.1 to discuss reionization of dark atoms by shock heating during structure formation.
Below eq. (10), added caveats concerning recent progress in identifying missing satellites, and the role of baryonic feedback.
Added references here.
At end of section 2.4, added discussion about the possible size of symmetric component of dark matter, and updated fig. 5 to illustrate this point.
Restored missing partial derivative symbols above eq. (26).
Below eq. (28), qualified the statement about ionized remnants being swept out of the galaxy by supernovae shocks, and added references.
Added brief discussion and reference to twin Higgs mirror sector at end of section 2.7.1.
Fixed (left)->(right) in caption of Fig. 10
Below eq. (31), added mention of MW satellite counts and reference, for constraining warm DM.
Fixed power of M in Fig. 12 caption.
Added discussion and references below eq. (37).
Below eq. (40), added clarification of pion decay constant definition.
Added footnote 7 and reference to discuss instability of DM candidate in SO(6)/SO(5) composite Higgs model.
Below eq. (48), fixed incorrect subscript.
Eq. (59), added PandaX-4T limit
Added acknowledgments.
Corrected typos found by student reader.
Added paragraph at end of section 2.1 to discuss reionization of dark atoms by shock heating during structure formation.
Below eq. (10), added caveats concerning recent progress in identifying missing satellites, and the role of baryonic feedback.
Added references here.
At end of section 2.4, added discussion about the possible size of symmetric component of dark matter, and updated fig. 5 to illustrate this point.
Restored missing partial derivative symbols above eq. (26).
Below eq. (28), qualified the statement about ionized remnants being swept out of the galaxy by supernovae shocks, and added references.
Added brief discussion and reference to twin Higgs mirror sector at end of section 2.7.1.
Fixed (left)->(right) in caption of Fig. 10
Below eq. (31), added mention of MW satellite counts and reference, for constraining warm DM.
Fixed power of M in Fig. 12 caption.
Added discussion and references below eq. (37).
Below eq. (40), added clarification of pion decay constant definition.
Added footnote 7 and reference to discuss instability of DM candidate in SO(6)/SO(5) composite Higgs model.
Below eq. (48), fixed incorrect subscript.
Eq. (59), added PandaX-4T limit
Added acknowledgments.
Corrected typos found by student reader.

---

## Editorial Decision

published